# KL Penalty Control via Perturbation for Direct Preference Optimization

**Sangkyu Lee**[1,*]    **Janghoon Han**[2]    **Hosung Song**[2]
**Stanley Jungkyu Choi**[2]    **Honglak Lee**[2,3]    **Youngjae Yu**[4]
Yonsei University[1]    LG AI Research[2]
University of Michigan, Ann Arbor[3]    Seoul National University[4]
oddqueue@yonsei.ac.kr    youngjaeyu@snu.ac.kr

## Abstract

Direct Preference Optimization (DPO) demonstrates the advantage of aligning a large language model with human preference using only an offline dataset. However, DPO has the limitation that the KL penalty, which prevents excessive deviation from the reference model, is static throughout the training process. Several methods claim to change this static KL penalty of DPO into a dynamic one, but no approach can adaptively assign different KL penalties for each preference pair. In this paper, we propose $\varepsilon$-Direct Preference Optimization ($\varepsilon$-DPO), which allows adaptive control of the KL penalty strength $\beta$ for each preference pair. Specifically, $\varepsilon$-DPO adaptively controls $\beta$ for each preference pair based on the monotonicity of logits as a preference model under the perturbation of $\beta$ during training. This is equivalent to adjusting the KL penalty by checking whether the change in training-time temperature can lead to better preference confidence as preference models by simply reusing the logit of the current policy and the reference policy. Experimental results show that the simple criterion of $\varepsilon$-DPO for KL penalty relaxation significantly improves DPO compared to most existing direct alignment algorithms on general chatbot benchmarks and reveal that this KL penalty control criterion can reflect confusion as a preference model and provide an efficient KL trade-off, highlighting the significance of instance-level adaptive KL penalty control in DPO.[1]

## 1 Introduction

Aligning large language models with human preferences for helpfulness and harmless principles [2, 4, 9] is a crucial requirement for general chatbot agents. Reinforcement Learning from Human Feedback (RLHF) [46] is the pioneering approach that regards the alignment of large language models as a reward maximization problem and solves it by reinforcement learning [37]. However, the complicated training pipeline of RLHF increases the training complexity and computation cost of the rollout for online reinforcement learning, in addition to the difficulty of collecting human preference datasets. Moreover, introducing a trained reward model as a proxy reward function to replace the intractable ground-truth human preference reward function makes large language models suffer from the side effect of reward over-optimization [13] inherited from the reward models.

Direct Preference Optimization (DPO) [32] proposes an alternative approach to reform the limitation of RLHF by converting the policy optimization problem into a preference modeling problem and performing alignment using only the offline preference dataset. It shows comparable performance

---

* Work done during internship at LG AI Research.

[1]The code is available at github.com/oddqueue/e-dpo.

**1) Monotonically Decreasing** $(x, y^w, y^l) \Rightarrow \beta \to \beta_\varepsilon^-$

$$\log \frac{\pi_{\hat\theta(\beta_\varepsilon^-)}(y^w|x)}{\pi_{\hat\theta(\beta_\varepsilon^-)}(y^l|x)} \quad \log \frac{\pi_{\theta(\beta)}(y^w|x)}{\pi_{\theta(\beta)}(y^l|x)} \quad \log \frac{\pi_{\hat\theta(\beta_\varepsilon^+)}(y^w|x)}{\pi_{\hat\theta(\beta_\varepsilon^+)}(y^l|x)}$$

**2) Monotonically Increasing** $(x, y^w, y^l) \Rightarrow \beta \to \beta_\varepsilon^+$

$$\log \frac{\pi_{\hat\theta(\beta_\varepsilon^-)}(y^w|x)}{\pi_{\hat\theta(\beta_\varepsilon^-)}(y^l|x)} \quad \log \frac{\pi_{\theta(\beta)}(y^w|x)}{\pi_{\theta(\beta)}(y^l|x)} \quad \log \frac{\pi_{\hat\theta(\beta_\varepsilon^+)}(y^w|x)}{\pi_{\hat\theta(\beta_\varepsilon^+)}(y^l|x)}$$

Figure 1: $\varepsilon$-DPO adaptively controls $\beta$ corresponding to the KL penalty strength for each preference pair by checking monotonicity of the log-likelihood ratio of the chosen response and the rejected according to perturbation of training-time $\beta$ by estimating the perturbed policies by reusing logits.

while skipping the reward modeling process required by RLHF and has become an effective alternative approach for alignment. In particular, subsequent studies with various modifications to the DPO objective function open a new research domain called direct alignment algorithms [33], which perform alignment directly from offline preference datasets without training separate reward models.

However, DPO assumes that $\beta$ and the reference policy, which define a KL penalty that prevents excessive deviations from the reference model in RLHF, are fixed for exploiting the existence of a closed-form solution derived from the objective function of the RLHF. However, this assumption can lead to suboptimal results, since the KL penalty can be regarded as a Lagrangian relaxation of the constraint optimization defined by the trust region [37]. In this regard, $\beta$-DPO [41] argues that $\beta$ should be adaptively chosen according to the quality of the preference pair but fails to control $\beta$ at the instance-level and proposes a batch-level control method. On the other hand, TR-DPO [14] claims to periodically update the reference policy to reduce over-optimization [33], but it may induce unnecessary KL divergence for improvement since the update is not adaptive.

In this paper, we present $\varepsilon$-**Direct Preference Optimization** ($\varepsilon$-**DPO**), a simple instance-level adaptive KL penalty control for DPO that neither TR-DPO nor $\beta$-DPO achieves. Specifically, we check the advantage of adjusting $\beta$ for each preference pair by observing the monotonicity of the log-likelihood ratio between the chosen response and the rejected response if the $\beta$ used during training was perturbed, as described in Figure 1. Here, the criterion for controlling $\beta$ does not require batch-level statistics, and the policy under the perturbed $\beta$ can be estimated by reusing the logits from the policy and reference policy. This criterion results in independence from the choice of micro-batch size and no additional computation requirements for model updates, unlike $\beta$-DPO and TR-DPO.

Experimental results demonstrate that the instance-level adaptive criterion of $\varepsilon$-DPO remarkably improves DPO, better than $\beta$-DPO and TR-DPO, to outperform most direct alignment algorithms that modify the DPO objective function [43, 45, 3, 42, 12, 18, 31, 29]. This reveals that the static KL penalty of DPO is the major bottleneck to final model performance and highlights the importance of instance-level adaptive KL penalty control. Furthermore, we confirm that the variation of $\beta$ determined by the adaptive criterion in $\varepsilon$-DPO reflects the confusion as a preference model, which is not addressed in the adaptive $\beta$ control criterion proposed by $\beta$-DPO. We also find that the adaptive KL penalty control of $\varepsilon$-DPO is crucial for an efficient KL trade-off compared to TR-DPO, which is not an adaptive KL penalty control because of the periodic update of the reference policy.

Our contribution to the alignment of large language models can be summarized as threefold: (1) We present $\varepsilon$-DPO that changes the static KL penalty of DPO and adaptively controls the KL penalty at the instance-level with a simple criterion, which can outperform most direct alignment algorithms proposed as alternatives to DPO in general chatbot benchmarks. (2) We show that DPO's perspective, which reparameterizes the policy to a preference model, can be converted to an approach for controlling the KL penalty at the instance-level by estimating the preference confidence changes according to the perturbation of $\beta$ used during training. (3) We demonstrate that this instance-level adaptive KL penalty control distinguishes confusing preference pairs and achieves an efficient KL trade-off, and neither is addressed in existing research that performs KL penalty relaxation.

## 2 Preliminaries

**Reinforcement Learning from Human Feedback**   To obtain a language model that aligns with human preference, RLHF [46] introduces reinforcement learning. It is equivalent to approaching preference alignment as a reward maximization problem, where we find a policy $\pi$ that maximizes a ground-truth reward function $r^*$ representing a human preference score for a response $y$ obtained

from a corresponding policy for a given prompt $x$. However, since the ground-truth reward function cannot be accessed, a reward model trained from the preference dataset is introduced as a proxy reward function. On the other hand, to prevent the policy update from deviating too much from the current policy from the initial policy, the KL divergence from the reference policy $\pi_{\text{ref}}$ serves as a penalty and regards the initial policy as a reference policy. At this time, the coefficient $\beta$ controls the strength of the penalty. The optimal policy that satisfies the maximization of the modified objective function under $\beta$ has a closed-form solution $\pi_\beta^*$ with an intractable normalizing constant $Z_\beta^*$,

$$\pi_\beta^*(y|x) := \arg \max_\pi \{\mathbb{E}_{x,y}[r^*(x,y)] - \beta \mathbb{D}_{\text{KL}}(\pi||\pi_{\text{ref}})\}$$

$$= \frac{1}{Z_\beta^*(x)} \pi_{\text{ref}}(y|x) \exp\left(\frac{1}{\beta} r^*(x,y)\right),$$

$$\text{where } Z_\beta^*(x) = \sum_y \pi_{\text{ref}}(y|x) \exp\left(\frac{1}{\beta} r^*(x,y)\right).$$

**Direct Preference Optimization**   RLHF has a limitation in efficiency due to the additional training step of the reward model. In this respect, DPO [32] proposes an approach that can perform preference alignment without training the reward model. DPO focuses on the fact that the ground-truth reward function can be implicitly reparameterized by the closed-form solution $\pi_\beta^*$ and reference policy $\pi_{\text{ref}}$. If we assume the Bradley-Terry model [6] for the ground-truth human preference function, then the human preference can be modeled by the margin between the reward of the chosen response $y^w$ and the rejected response $y^l$ with the sigmoid function $\sigma$, which can cancel out the intractable term $Z_\beta^*$. From this observation, DPO performs preference alignment through preference model optimization using an offline dataset in the sense that obtaining an optimal policy through policy optimization in RLHF can be obtained by training a preference model given by the implicit reward $r_{\theta,\beta}$,

$$\mathcal{L}_{\text{DPO}}(x, y^w, y^l; \theta, \beta) := -\log \mathbb{P}_{\theta,\beta}(y^w \succ y^l|x),$$

$$\text{where } r_{\theta,\beta}(x,y) := \beta \log \frac{\pi_\theta(y|x)}{\pi_{\text{ref}}(y|x)} + Z_\beta(x;\theta),$$

$$\mathbb{P}_{\theta,\beta}(y^w \succ y^l|x) := \sigma\left(r_{\theta,\beta}(x,y^w) - r_{\theta,\beta}(x,y^l)\right).$$

## 3   $\varepsilon$-Direct Preference Optimization

In this section, we describe our proposed method, $\varepsilon$-**Direct Preference Optimization** ($\varepsilon$-**DPO**), that adaptively controls the KL penalty coefficient $\beta$ at the instance-level based on the logit monotonicity as a preference model according to the perturbation of $\beta$. Figure 2 illustrates the difference between $\varepsilon$-DPO and existing KL penalty relaxation methods for DPO, $\beta$-DPO [41] and TR-DPO [14].

### 3.1   Relaxation of the KL Penalty in DPO

The KL penalty introduced by RLHF can be regarded as an approach to solve the constrained optimization problem in the trust region [36] defined near the reference policy $\pi_{\text{ref}}$ as an unconstrained optimization by treating $\beta$ as a Lagrange multiplier [37]. From this perspective, even though DPO reformulates the problem of finding an optimal policy under fixed $\pi_{\text{ref}}$ and $\beta$ as a preference modeling problem, using a single $\beta$ and a fixed trust region for all instances may lead to suboptimal results. This hypothesis regarding relaxation of KL penalty can be supported by the experimental results of $\beta$-DPO [41] that adaptively control $\beta$ based on the statistics of implicit reward margin during the training process and TR-DPO [14] that updates $\pi_{\text{ref}}$ during the training process for preventing over-optimization [33] from the vanishing curvature of the loss landscape.

However, $\beta$-DPO fails to perform instance-level $\beta$ control despite claiming that the quality of each preference pair should determine $\beta$. Instead, it performs batch-level $\beta$ control using momentum-based estimation of batch-level margin disparities, which is strongly affected by the micro-batch size. In addition, TR-DPO updates the reference model without adaptive criteria, which can lead to inefficient KL divergence trade-off between performance and incur computational costs for updating the reference model. Therefore, instance-level adaptive KL penalty control without requiring additional computational cost that achieves an efficient KL trade-off is still undiscovered for DPO.

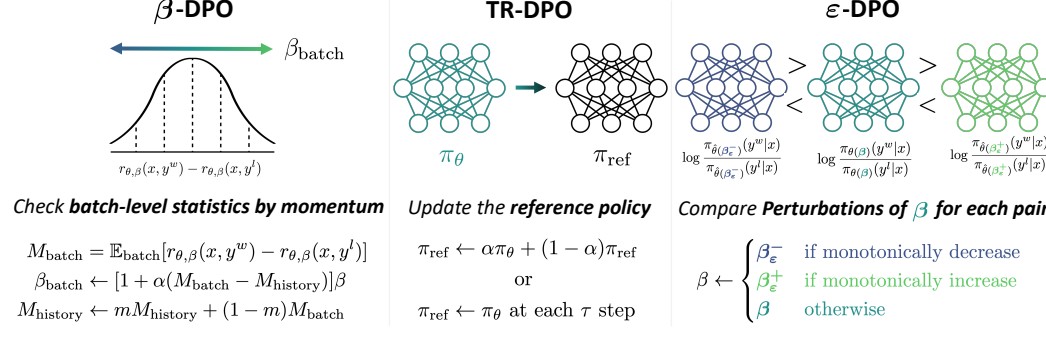

Figure 2: Comparison between $\varepsilon$-DPO and existing KL penalty relaxation methods for DPO, $\beta$-DPO [41] and TR-DPO [14]. Only $\varepsilon$-DPO achieves instance-level KL penalty relaxation compared to other methods, which control $\beta$ at batch-level or update the reference policy periodically.

## 3.2 Logit Monotonicity Under the KL Penalty Perturbation

Establishing a criterion for adaptively changing the KL penalty for each instance in the preference dataset is not a trivial problem. As a proxy criterion, we can revisit the DPO's assumption on the ground-truth preference model, the Bradley-Terry model [6], which assumes that the reward difference between two candidates imposes a total ordering. Formally, the policy obtained via DPO can function as a preference model $\mathbb{P}_{\theta,\beta}$ which can be expressed as a binary classifier,

$$\mathbb{P}_{\theta,\beta}(y^w \succ y^l|x) = \sigma\Big(\beta\big(z_\theta(x, y^w, y^l) - \gamma(x, y^w, y^l)\big)\Big),$$

by regarding the log-likelihood ratio between the chosen response and the rejected response from a preference triplet $(x, y^w, y^l) \in \mathcal{D}$ as a logit $z_\theta(x, y^w, y^l)$ and adaptive margin $\gamma(x, y^w, y^l)$,

$$z_\theta(x, y^w, y^l) := \log \frac{\pi_\theta(y^w|x)}{\pi_\theta(y^l|x)}, \ \ \gamma(x, y^w, y^l) := \log \frac{\pi_{\text{ref}}(y^w|x)}{\pi_{\text{ref}}(y^l|x)}.$$

This reveals that the KL penalty coefficient $\beta$ also serves as an inverse temperature of a binary classifier. For a given $\beta$, we define $\beta_\varepsilon^-$ and $\beta_\varepsilon^+$ with a positive constant $\varepsilon > 0$. That is, $\beta_\varepsilon^-$ and $\beta_\varepsilon^+$ refer to values that have been *perturbed* to be slightly larger or slightly smaller than the $\beta$,

$$\beta_\varepsilon^- := \frac{\beta}{1 + \varepsilon}, \ \ \beta_\varepsilon^+ := \frac{\beta}{1 - \varepsilon}.$$

Let us denote the parameters obtained via DPO as a function of $\beta$, $\theta(\beta) : \mathbb{R}^+ \to \Theta$. Consider the case we observe the strict *monotonicity* of logits happens according to the perturbation of $\beta$ on $\theta(\beta)$,

$$z_{\theta(\beta_\varepsilon^-)}(x, y^w, y^l) > z_{\theta(\beta)}(x, y^w, y^l) > z_{\theta(\beta_\varepsilon^+)}(x, y^w, y^l), \tag{1}$$

$$z_{\theta(\beta_\varepsilon^-)}(x, y^w, y^l) < z_{\theta(\beta)}(x, y^w, y^l) < z_{\theta(\beta_\varepsilon^+)}(x, y^w, y^l). \tag{2}$$

Suppose we assume that the observation corresponds to the hard label in the ground-truth preference model (i.e. $\mathbb{P}(y^w \succ y^l|x) = 1$). In this case, the preference model with a larger logit is considered more accurate under the Bradley-Terry model. Alternatively, it is equivalent to checking monotonic changes in preference confidence under the perturbation of training-time inverse temperature $\beta$ by whether the better separation of $y^w$ and $y^l$ can be obtainable at the same test-time temperature scaling [15] in the neighborhood of $\frac{1}{\beta}$. From this criterion, we can estimate the direction of adjusting $\beta$ for each instance within the neighborhood defined by $\varepsilon$ to increase preference confidence.

## 3.3 Estimating Policies Under the KL Penalty Perturbation

Note that $\theta(\beta)$ is an intractable function since it is equivalent to having access to models trained on each $\beta$ in the definition. However, Liu et al. [28] shows that optimal policy under $\frac{\beta}{\lambda}$ can be expressed through $\pi_\beta^*$ by re-weighting with importance ratio using $\pi_{\text{ref}}$. If we assume the autoregressive prior of optimal policy, then the optimal policy under $\frac{\beta}{\lambda}$ can be estimated by the optimal policy under $\beta$ and the reference policy, as we respecify the observation of Liu et al. [28] as Proposition 1,

**Proposition 1 (Liu et al. [28])** *Under the assumption of optimal autoregressive policy $\pi^*$ where the prompt $x \in \mathcal{X}$, response vocabulary $y_i \in \mathcal{V}$, and logit $f : \mathcal{X} \times \mathcal{V}^{i-1} \to \mathbb{R}^{|\mathcal{V}|}$, the optimal policy $\pi^*_{\frac{\beta}{\lambda}}$ can be approximated by the arithmetic mean of logits between $\pi^*_\beta$ and reference policy $\pi_{\mathrm{ref}}$,*

$$
\begin{aligned}
\pi^*_{\frac{\beta}{\lambda}}(y_{1:n}|x) &= \prod_{i=1}^{n} \pi^*_{\frac{\beta}{\lambda}}(y_i|x, y_{1:i-1}) \\
&\approx \prod_{i=1}^{n} \mathrm{Softmax}\big(\lambda f^*_\beta(x, y_{1:i-1}) + (1-\lambda) f_{\mathrm{ref}}(x, y_{1:i-1})\big)_{y_i}.
\end{aligned}
$$

*Proof. See Appendix A.*

Using Proposition 1, we can approximate $\pi_{\theta(\beta_\varepsilon^-)}$ and $\pi_{\theta(\beta_\varepsilon^+)}$ by trained policy and reference policy without accessing $\theta(\beta)$ since they are the approximated policies for $\pi^*_{\beta_\varepsilon^-}$ and $\pi^*_{\beta_\varepsilon^+}$. To adaptively control $\beta$ for each preference triplet $(x, y^w, y^l)$ during the training process, we regard the policy $\pi_\theta$ obtained in the current training step as the our best approximation of the optimal policy defined under current $\beta$ and estimate $\pi_{\theta(\beta_\varepsilon^-)}$ and $\pi_{\theta(\beta_\varepsilon^+)}$ for approximating intractable $z_{\theta(\beta_\varepsilon^-)}$ and $z_{\theta(\beta_\varepsilon^+)}$,

$$
\begin{aligned}
\pi_{\theta(\beta_\varepsilon^-)}(y_{1:n}|x) &\approx \prod_{i=1}^{n} \pi^*_{\beta_\varepsilon^-}(y_i|x, y_{1:i-1}) = \prod_{i=1}^{n} \pi^*_{\frac{\beta}{1+\varepsilon}}(y_i|x, y_{1:i-1}) \\
&\approx \prod_{i=1}^{n} \mathrm{Softmax}\big((1+\varepsilon) f_\theta(x, y_{1:i-1}) - \varepsilon f_{\mathrm{ref}}(x, y_{1:i-1})\big)_{y_i},
\end{aligned}
\tag{3}
$$

$$
\begin{aligned}
\pi_{\theta(\beta_\varepsilon^+)}(y_{1:n}|x) &\approx \prod_{i=1}^{n} \pi^*_{\beta_\varepsilon^+}(y_i|x, y_{1:i-1}) = \prod_{i=1}^{n} \pi^*_{\frac{\beta}{1-\varepsilon}}(y_i|x, y_{1:i-1}) \\
&\approx \prod_{i=1}^{n} \mathrm{Softmax}\big((1-\varepsilon) f_\theta(x, y_{1:i-1}) + \varepsilon f_{\mathrm{ref}}(x, y_{1:i-1})\big)_{y_i}.
\end{aligned}
\tag{4}
$$

Recall that we need not only the logit of the current policy $f_\theta$ but also the logit of the reference policy $f_{\mathrm{ref}}$ to compute the estimated log-likelihood ratio. However, in order to compute the loss function of DPO, $\mathcal{L}_{\mathrm{DPO}}$, the log-likelihood from the reference policy must be computed for each training instance, which allows us to simply reuse $f_{\mathrm{ref}}$ for estimation without any additional computation cost of model forward passes. Therefore, we determine the $\tilde{\beta}$, which is used for the KL penalty coefficient in the current training step for each training preference triple instance $(x, y^w, y^l)$,

$$
\tilde{\beta}(x, y^w, y^l; \theta) = \begin{cases} \beta_\varepsilon^- & \text{if (1),} \\ \beta_\varepsilon^+ & \text{if (2),} \\ \beta & \text{otherwise.} \end{cases}
\tag{5}
$$

After the model update, the $\beta$, which corresponds to the optimal policy that the current policy targets, should be changed depending on $\tilde{\beta}$ used in $\mathcal{L}_{\mathrm{DPO}}$ for each instance. Therefore, we need to modify the baseline $\beta$ for the next training step, and we simply update the $\beta$ with the mean statistics of $\tilde{\beta}$ determined across the batch used in the update. Note that $\tilde{\beta}$ is determined independently of the batch-level statistic, so the adaptive control of $\beta$ in $\varepsilon$-DPO can be performed independently of the choice of micro-batch size. Algorithm 1 summarizes the entire training process of $\varepsilon$-DPO.

---

**Algorithm 1** $\varepsilon$-Direct Preference Optimization

---

**Require:** policy $\pi_\theta$, reference policy $\pi_{\mathrm{ref}}$, initial KL penalty coefficient $\beta$, and perturbation size $\varepsilon$
1: **while** not converged **do**
2:      Sample training batch of preference triplet $(x, y^w, y^l) \sim \mathcal{D}$.
3:      Estimate the policies under the perturbation $\pi_{\hat{\theta}(\beta_\varepsilon^-)}$ and $\pi_{\hat{\theta}(\beta_\varepsilon^+)}$ according to 3 and 4.
4:      Determine instance-level KL penalty coefficients $\tilde{\beta}(x, y^w, y^l; \theta)$ according to 5.
5:      Update $\pi_\theta$ by $\mathcal{L}_{\mathrm{DPO}}$ with $\tilde{\beta}(x, y^w, y^l; \theta)$ and then $\beta \leftarrow \mathbb{E}_{x, y^w, y^l}[\tilde{\beta}(x, y^w, y^l; \theta)]$.
6: **end while**
7: **return** aligned policy $\pi_\theta$.

---

Table 1: AlpacaEval 2 [11], Arena-Hard [25], and MT-Bench [21] results of the `Instruct` setting proposed by SimPO [29]. LC and WR denote length-controlled win rate and win rate. Results of other direct alignment algorithms [32, 43, 45, 3, 42, 12, 18, 31] are from the official paper of SimPO.

| Method | Mistral-Instruct (7B) | | | | Llama-3-Instruct (8B) | | | |
| | AlpacaEval 2 | | Arena-Hard | MT-Bench | AlpacaEval 2 | | Arena-Hard | MT-Bench |
| | LC (%) | WR (%) | WR (%) | Score (1-10) | LC (%) | WR (%) | WR (%) | Score (1-10) |
|---|---|---|---|---|---|---|---|---|
| SFT | 17.1 | 14.7 | 12.6 | 7.5 | 26.0 | 25.3 | 22.3 | 8.1 |
| DPO | 26.8 | 24.9 | 16.3 | 7.6 | 40.3 | 37.9 | 32.6 | 8.0 |
| RRHF | 25.3 | 24.8 | 18.1 | 7.6 | 31.3 | 28.4 | 26.5 | 7.9 |
| SLiC-HF | 24.1 | 24.6 | 18.9 | **7.8** | 26.9 | 27.5 | 26.2 | 8.1 |
| IPO | 20.3 | 20.3 | 16.2 | **7.8** | 35.6 | 35.6 | 30.5 | **8.3** |
| CPO | 23.8 | 28.8 | **22.6** | 7.5 | 28.9 | 32.2 | 28.8 | 8.0 |
| KTO | 24.5 | 23.6 | 17.9 | 7.7 | 33.1 | 31.8 | 26.4 | 8.2 |
| ORPO | 24.5 | 24.9 | 20.8 | 7.7 | 28.5 | 27.4 | 25.8 | 8.0 |
| R-DPO | 27.3 | 24.5 | 16.1 | 7.5 | 41.1 | 37.8 | 33.1 | 8.0 |
| SimPO | 32.1 | **34.8** | 21.0 | 7.6 | 44.7 | 40.5 | 33.8 | 8.0 |
| $\varepsilon$-DPO | **35.6** | 29.6 | 17.2 | **7.8** | **46.4** | **44.9** | **36.7** | 8.0 |

# 4 Experiments

In this section, we conduct experiments to validate the $\varepsilon$-DPO. We mainly check the feasibility of $\varepsilon$-DPO for general chatbot alignment using UltraFeedback [9], compared to the direct alignment algorithms [32, 43, 45, 3, 42, 12, 18, 31, 29]. We also use Anthropic-HH [4] for analyzing the proposed adaptive KL penalty control and comparing with existing methods for KL penalty relaxation of DPO [41, 14]. The implementation details for each experimental setting are in Appendix B.

## 4.1 Datasets and Evaluations

**UltraFeedback**    UltraFeedback [9] is an AI feedback dataset where GPT-4 [1] rates responses obtained from four different language models. We strictly follow the experimental setting proposed by SimPO [29], which conducts broad range of hyperparameter search then comparing best performance of various direct alignment algorithms including DPO [32], RRHF [43], SLiC-HF [45], IPO [3], CPO [42], KTO [12], ORPO [18], and R-DPO [31] for robust comparison due to the hyperparamter sensitivity of direct alignment algorithms. Specifically, we use the `Instruct` setting starting from `Mistral-7B-Instruct-v0.2` [20] and `Meta-Llama-3-8B-Instruct` [10]. We evaluate resulting models by general chatbot benchmarks, AlpacaEval 2 [11], Arena-Hard [25], and MT-Bench [21].

**Anthropic-HH**    Anthropic-HH [4] is a human preference dialogue dataset containing two subsets based on the helpfulness and harmlessness principle. Here, we use `helpful-base` and `harmless-base` splits to validate the criterion using logit monotonicity for instance-level $\beta$ control used in $\varepsilon$-DPO and the efficiency in terms of trade-off between performance and KL divergence [33]. We choose `gemma-2-2B` [39] to obtain the reference policy through Supervised Fine-tuning with chosen responses. Following DPO [32], we evaluate the models trained with each method under various $\beta$ in the single-turn dialogue setting. We regard PairRM [21] as an external evaluator for checking performance by win rate, comparing their responses and chosen responses in the test splits.

## 4.2 Experimental Results on UltraFeedback

**Overall Performance of $\varepsilon$-DPO**    In Table 1, we observe that $\varepsilon$-DPO shows notable performances compared to DPO across AlpacaEval 2 [11], Arena-Hard [25], and MT-Bench [21]. In particular, we find that the performance of $\varepsilon$-DPO outperforms most direct alignment algorithms, which generally modify the loss function, highlighting that the major assumption of fixed KL penalty in DPO is overlooked. Simultaneously, we observe that $\varepsilon$-DPO performs better than other KL penalty relaxation approaches [41, 14] from Table 2. We further consider an experimental setting

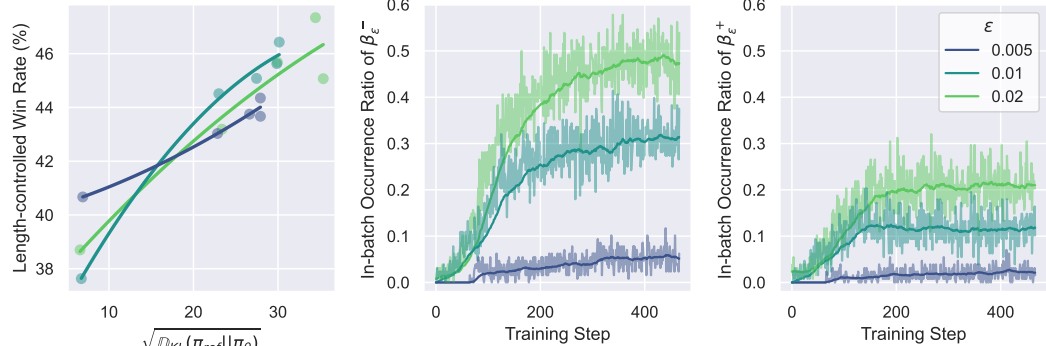

Figure 3: Intra-epoch training dynamics of `Llama-3-Instruct` according to the change of $\varepsilon$. We additionally plot the fitted curves of AlpacaEval 2 LC results of each checkpoint and exponential moving average lines of the in-batch occurrence ratio on $\beta_\varepsilon^-$ and $\beta_\varepsilon^+$ for better visual representation.

that uses `Qwen2.5-7B-Instruct` [40] as the base model, extending the original SimPO's experimental setting. Specifically, we train the base model with the best hyperparameters obtained in the `Llama-3-Instruct` setting, without performing a hyperparameter search, as shown in Table 3. Interestingly, $\varepsilon$-DPO outperforms DPO even with hyperparameters obtained in the `Llama-3-Instruct` setting, whereas SimPO is inferior to DPO despite SimPO showing comparable performance in the `Llama-3-Instruct`. We speculate that introducing a fixed margin as a hyperparameter to mitigate the KL penalty, as in SimPO, can be an effective approach only if adequate hyperparameter search precedes it; adherence to the reference policy remains crucial otherwise. In addition to general chatbot benchmarks for preference alignment, we also check the Huggingface Open LLM Leaderboard [5] to see the impact $\varepsilon$-DPO on specific downstream tasks in Appendix C, but we find $\varepsilon$-DPO also follows the general trend of direct alignment algorithms [29]. Thus, we can confirm that instance-level KL penalty control significantly impacts the performance of general chatbot agents.

**Influence of $\varepsilon$ to the Training Dynamics** In $\varepsilon$-DPO, the perturbation scale $\varepsilon$ is used for checking logit monotonicity as a preference model in the neighborhood of the current $\beta$, for estimating policies under perturbation of the KL penalty. Therefore, it can be chosen within a reasonable range to estimate the approximated policies corresponding to $\beta_\varepsilon^-$ and $\beta_\varepsilon^+$. However, $\varepsilon$ can influence training dynamics since $\varepsilon$ determines the scale of the instance-level KL penalty coefficient $\tilde{\beta}$. We further analyze the intra-epoch training dynamics on `Llama-3-Instruct` settings according to $\varepsilon$. We compare the forward KL divergence $\mathbb{D}_{KL}(\pi_{ref}||\pi_\theta)$ [33] and performance on AlpacaEval 2 using checkpoints obtained at 0.2 intervals during the training, along with the in-batch occurrence ratio of $\beta_\varepsilon^-$ and $\beta_\varepsilon^+$, as shown in Figure 3. We find that adaptive control occurs more frequently for both $\beta_\varepsilon^-$ and $\beta_\varepsilon^+$ as $\varepsilon$ increases, leading to an acceleration of the increase of the KL divergence and performance. We also observe that the performance at the beginning of training tends to be lower when $\varepsilon$ is higher. We speculate that the trained policy at the beginning of training is insufficient to estimate the optimal policy, making the approximation unstable at the high $\varepsilon$ level.

Table 2: AlpacaEval 2 and Arena-Hard results of $\beta$-DPO [41], TR-DPO [14] from each official papers compared to $\varepsilon$-DPO in the `Llama-3-Instruct`.

| Method | AlpacaEval 2 | | Arena-Hard |
|---|---|---|---|
| | LC (%) | WR (%) | WR (%) |
| SFT | 26.0 | 25.3 | 22.3 |
| DPO | 40.3 | 37.9 | 32.6 |
| $\beta$-DPO | 43.4 | 38.2 | - |
| TR-DPO$^\tau$ | 42.8 | **47.2** | 32.4 |
| TR-DPO$^\alpha$ | 43.5 | 46.8 | 34.7 |
| $\varepsilon$-DPO | **46.4** | 44.9 | **36.7** |

Table 3: AlpacaEval 2, Arena-Hard and MT-Bench results of `Qwen2.5-7B-Instruct` with best hyperparameters found in the `Llama-3-Instruct`.

| Method | AlpacaEval 2 | | Arena-Hard | MT-Bench |
|---|---|---|---|---|
| | LC (%) | WR (%) | WR (%) | Score (1-10) |
| SFT | 27.8 | 27.9 | 51.8 | 8.6 |
| DPO | 41.6 | **46.3** | 66.8 | 8.9 |
| SimPO | 32.4 | 46.0 | 60.2 | 8.8 |
| $\varepsilon$-DPO | **42.5** | 46.1 | **67.5** | **9.1** |

**Analysis of Computation Cost** Although the instance-level KL penalty control of $\varepsilon$-DPO improves performance over DPO and incurs no additional model forward passes cost, it incurs additional computation costs for estimating policies with the training-time $\beta$ perturbation, which warrants further analysis. Formally, the estimated forward passes cost $C_f$ and backward passes cost $C_b$ per token in FLOPs, following $C_f \approx 2N$ and $C_b \approx 2C_f$ for a given model parameter size $N$, excluding the embedding

Table 4: Wall-time increment $\Delta t$ of $\varepsilon$-DPO during the training of `Instruct` setting. For the step-level result, we report the average wall-time increment measured during a single training epoch.

| $\Delta t$ | Mistral-Instruct | Llama-3-Instruct |
|---|---|---|
| Step (sec) | 0.0008 | 0.0006 |
| Epoch (sec) | 0.3808 | 0.3002 |
| Ratio (%) | 0.0064 | 0.0045 |

layer [23]. In the case of DPO, since forward and backward passes for the policy model and forward pass for the reference model occur, the FLOPs per token can be approximated as $8N$. When we approximate the policy model under perturbation of $\beta$, $(2v + v + 5v)$ FLOPs are added per token for a given vocabulary size $v$, which corresponds to two scalar-vector multiplications, vector addition, and log-softmax operation, respectively. This implies that the relative ratio of additional computation cost in FLOPs per token compared to the computation cost of DPO can be roughly approximated as $\frac{2v}{N}$; therefore, because $v \ll N$ in general, the additional computation cost required by $\varepsilon$-DPO is negligible. To verify whether $\varepsilon$-DPO follows such a small computational cost empirically, we compared the wall-time increment $\Delta t$ during training of $\varepsilon$-DPO compared to DPO under `Mistral-Instruct` and `Llama-3-Instruct` settings as Table 4, which confirms our computation analysis.

## 4.3 Experimental Results on Anthropic-HH

**Variants of KL Penalty Control Strategy** By default, $\varepsilon$-DPO shares the same $\varepsilon$ that defines the neighborhood to check logit monotonicity and determine the relaxation strength performed on $\beta$ in each instance. However, to understand how $\varepsilon$ affects the training dynamics, it is beneficial to compare the default strategy with alternative strategies that use different values of $\varepsilon$ for each case. We compare the default strategy with strategies that use different values of $\varepsilon_c$ and $\varepsilon_s$,

Table 5: Ablation of instance-level adaptive KL penalty control strategy of $\varepsilon$-DPO according to $\varepsilon$.

| WR / sgn($\varepsilon_i$) | | $\varepsilon_s$ | |
|---|---|---|---|
| (% / Avg) | 0.005 | 0.01 | 0.02 |
| $\varepsilon_c$ 0.005 | 76.4 / 0.07 | 76.7 / 0.07 | 76.4 / 0.07 |
|    0.01 | 78.4 / 0.24 | **79.2 / 0.25** | 77.4 / 0.24 |
|    0.02 | 74.9 / 0.34 | 74.2 / 0.35 | 74.6 / 0.34 |

which are used to check logit monotonicity and to define the step size of the KL penalty control, respectively, when $\beta$ is fixed at 0.05. Simultaneously, we also check the average occurrence of $\beta_\varepsilon^-$ and $\beta_\varepsilon^+$ through the corresponding average step direction for each instance sgn($\varepsilon_i$) in the batch to compare how the scale of $\varepsilon$ affects the KL penalty relaxation. Table 5 shows that using different $\varepsilon_c$ and $\varepsilon_s$ produces suboptimal result compared to the default strategy. Furthermore, we can see that the adaptive KL penalty control of $\varepsilon$-DPO is strongly influenced by $\varepsilon_c$ since a similar level of sgn($\varepsilon_i$) is observed for the same $\varepsilon_c$, and only imposing a higher $\varepsilon_c$ can lead to worse performance than DPO compared to $\varepsilon_s$. In the sense that adopting higher $\varepsilon$ for the estimating perturbed policy can be understood as the stronger extrapolation for approximation of distribution, we can see that it is necessary to set an appropriately small size of $\varepsilon$, as large $\varepsilon$ risks increasing the probability of making a wrong decision for KL penalty relaxation in terms of weaker approximation to the optimal policy.

**Behavior of Logit Monotonicity Criterion** $\beta$-DPO [41] chooses a higher $\beta$ for preference pairs with larger implicit reward margins to update the current policy conservatively from the reference policy. This is motivated by the claim that large implicit reward margins reflect higher quality gaps of response pairs corresponding to meaningless training signals. In this respect, we analyze the implicit reward margin of preference pairs where logit monotonicity according to the perturbation of $\beta$ happened in policies trained by DPO using Antropic-HH, as shown in Figure 4a. We find that $\varepsilon$-DPO performs opposite decisions compared to $\beta$-DPO, assigning a higher $\beta$ for preference pairs, revealing high confusion based on the observation that preference pairs with monotonically increasing logits show low confidence as a preference model. Also, this implies that $\varepsilon$-DPO reflects confusion on the preference label to the training signals by scaling the gradient of DPO loss through controlling $\beta$ [32]. Furthermore, we confirm that implicit reward margins do not always represent the quality of preference pairs through qualitative analysis in Appendix D. Therefore, we suspect that $\beta$-DPO fails

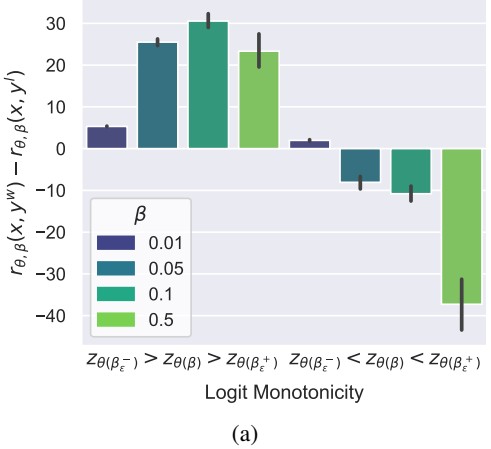
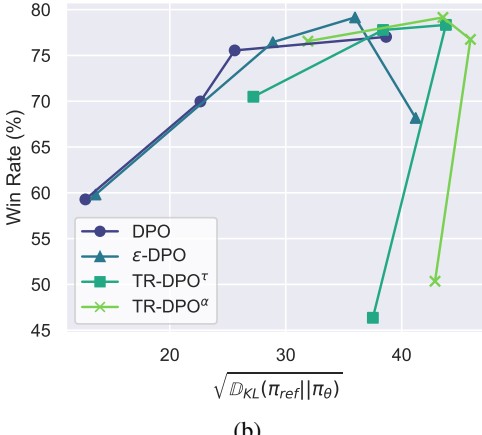

$$(a) \qquad\qquad\qquad\qquad (b)$$

Figure 4: (a) Implicit reward margin of pairs showing logit monotonicity in policies trained with DPO under various $\beta$. Each error bar indicates the 0.95 confidence interval. (b) Pareto frontier between KL divergence and win rate, which is measured by comparing with chosen responses in the test split.

on the instance-level KL penalty control because it strongly relies on the implicit reward margins that do not always represent the quality of preference pairs, so that it fails to detect confusing examples.

**KL Trade-off Efficiency of Adaptive Control**  As TR-DPO [14] claims, increasing the KL divergence would be desirable as a trade-off when deviating from the reference policy improves the performance. However, the over-optimization of direct alignment algorithms [33] emphasizes that it is necessary to check the Pareto frontier to determine whether performance improvements can be achieved without indiscriminately expanding the KL divergence because of degenerating behavior as the KL divergence grows. Figure 4b depicts the Pareto frontier between forward KL divergence and win rate compared with chosen responses in the test split, measured using Antropic-HH. Each model is trained through DPO, $\varepsilon$-DPO and two variants of TR-DPO, TR-DPO$^\tau$, which hard-updates the reference policy by the fixed interval, and TR-DPO$^\alpha$, which soft-updates the reference policy through weight merging, sharing the same $\beta$ range, [0.01, 0.05, 0.1, 0.5]. We can see that $\varepsilon$-DPO shows better performance than DPO, simultaneously achieving better KL trade-off efficiency than TR-DPO. Also, we can observe that regardless of the two variants, TR-DPO induces more KL divergence than DPO and $\varepsilon$-DPO and cannot achieve similar performance under the same KL budget as $\varepsilon$-DPO. This highlights the efficiency of $\varepsilon$-DPO in the KL trade-off and implies that controlling the KL penalty in a non-adaptive manner can induce excessive relaxation for performance improvements.

**Sensitivity of $\varepsilon$ on the Approximation**  The logit monotonicity criterion assumes two conditions: (1) The current policy can sufficiently approximate the optimal policy for a given $\beta$. (2) When observing logit monotonicity, the logit function $z_{\theta(\beta)}$ maintains monotonic order in the entire neighborhood $(\beta_\varepsilon^-, \beta_\varepsilon^+)$. However, these conditions can be significantly affected by the current policy and the choice of $\varepsilon$ during training. To verify how much these conditions can be satisfied, we additionally check the upper bound of $\varepsilon$ that defines a neighborhood of all values in the neighborhood consistently satisfies the logit monotonicity criterion for triplets $(x, y^w, y^l)$, through checkpoints obtained in 0.1 epoch intervals on DPO when $\beta$ is fixed as 0.05.

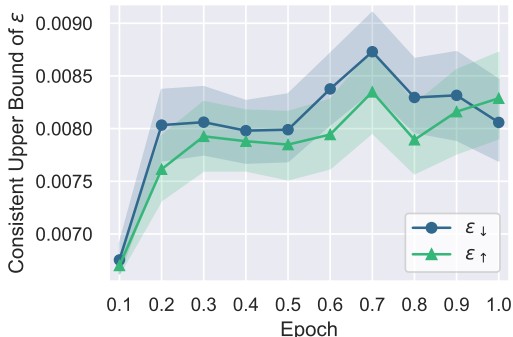

Figure 5: Changes of upper bound of $\varepsilon$ consistently satisfying the monotonically decreasing or increasing criterion with the 0.95 confidence band.

That is, assuming that the approximation of the optimal policy for the current $\beta$ improves with the increase in training steps, we verify the smoothness of logit monotonicity with respect to $\varepsilon$ by observing that the upper bound of $\varepsilon$ yields consistent decisions compared to smaller values. We test 100 uniform sample points of $\varepsilon$ over the range (0.005, 0.02). We observe that $\varepsilon_\downarrow$ and $\varepsilon_\uparrow$, which correspond to the

expected upper bound of $\varepsilon$ for monotonic decreasing and increasing logits, respectively, converged almost at 0.008 after 0.2 epochs, similar to the best result of previous experiments, as shown in Figure 5. Furthermore, the low value at 0.1 epoch is consistent with the phenomenon observed in the early stages of training in the `Llama-3-Instruct`. Therefore, we can confirm that relatively stable estimations of policy under perturbation of $\beta$, except for the early stage of training.

## 5    Related Works

**Direct Alignment Algorithms**    Many variants of direct alignment algorithms perform alignment on offline preference datasets without an external reward model. DPO [32] performs alignment through preference modeling with the implicit reward derived from the optimal policy of reward maximization under the KL divergence regularization. RRHF [43] performs alignment by training to maintain the likelihood margin between preference ranks. KTO [12] changes the assumptions of the Bradley-Terry model [6] used by DPO and introduces Prospect Theory [22], and IPO [3] converts to the root-finding problem for strengthening the KL constraint. SLiC-HF [45], CPO [42], ORPO [18], and SimPO [29] train without reference models by compensating the KL penalty through behavior cloning, margin loss, contrastive loss, odds ratio loss, and fixed margin by replacing the implicit rewards.

**Reward Over-optimization and KL Penalty**    Since RLHF [46] utilizes a trained reward model, it amplifies the limitations of the reward model as it is optimized toward an imperfect reward, according to Goodhart's Law [19], and this is called reward over-optimization [13]. However, Rafailov et al. [33] finds that direct alignment algorithms also experience similar reward over-optimization, regardless of the variant. Direct alignment algorithms commonly show humped curves of performance according to the increase of the KL divergence from the reference model during training. TR-DPO [14] argues that this is due to the Hessian of the loss landscape converging to zero as the implicit reward margin grows, so they update the reference model for mitigating this phenomenon. On the other hand, $\beta$-DPO [41], which also performs relaxation of the KL penalty, claims that adaptively changing $\beta$ through the statistics of the implicit reward margin is required to reflect the quality of the preference pair.

**Combining Sampling Distribution**    Combining sampling distributions can be utilized to estimate a new sampling distribution with specific characteristics. Contrastive Decoding [26] shows that the log-likelihood margins of the expert and amateur language models can enhance response diversity by penalizing incorrect response patterns favored by the amateur language model. Sanchez et al. [35] shows that classifier-free guidance [17] can enhance prompt relativity in language modeling by treating prompts as conditions. Mitchell et al. [30] estimates the importance ratio of the optimal distribution in RLHF by combining the change during instruction-tuning in a small language model with the large language model to approximate fine-tuning. Inspired by the theoretical motivation of Mitchell et al. [30], Liu et al. [28] shows that the sampling distribution of the policy trained by DPO with different $\beta$ can be approximated by importance sampling using the reference policy.

## 6    Conclusion

In this paper, we present $\varepsilon$-Direct Preference Optimization ($\varepsilon$-DPO), an instance-level adaptive KL penalty control method for DPO, adjusting the KL penalty coefficient $\beta$ by observing the monotonicity of the log-likelihood ratio between the chosen response and the rejected response when the $\beta$ used during training is perturbed. This simple criterion only requires estimating the policy under the perturbed $\beta$, which can be efficiently estimated by reusing the policy and reference policy logits without relying on batch-level statistics and requiring computation of reference policy updates. $\varepsilon$-DPO shows significantly better performance than DPO and also surpasses most existing direct alignment algorithms in general chatbot benchmarks. In particular, the criterion of $\varepsilon$-DPO shows a more efficient KL trade-off than the non-adaptive KL penalty relaxation while reflecting the confusion on preference pairs, emphasizing the importance of an appropriate instance-level KL penalty relaxation.

**Limitations**    $\varepsilon$-DPO requires the reference policy because it has a KL penalty from the reference policy, like DPO in default. It leads to requirements of additional memory consumption and computation for the reference policy compared to other direct alignment algorithms that do not perform regularization through the reference policy [45, 42, 18, 29]. Still, $\varepsilon$-DPO can reduce additional resources by pre-computing the logits of the responses from the reference policy, similar to DPO.

## Acknowledgements

This work was supported by LG AI Research. This work was supported by the Institute of Information & Communications Technology Planning & Evaluation (IITP) grants funded by the Korea government (MSIT) (No. RS-2025-II211343, Artificial Intelligence Graduate School Program (Seoul National University)). This work was supported by the National Research Foundation of Korea (NRF) grants funded by the Korean government (MSIT) (Nos. RS-2024-00354218 and RS-2024-00353125). This work was supported by the Institute of Information & Communications Technology Planning & Evaluation (IITP) grants funded by the Korea government (MSIT) (No. RS-2025-02263598, Development of Self-Evolving Embodied AGI Platform Technology through Real-World Experience). This work was supported by an IITP grant funded by the Korean Government (MSIT) (No. RS-2024-00353131).

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

# A   Proof of Proposition 1

**Proposition 1 (Liu et al. [28])** *Under the assumption of optimal autoregressive policy $\pi^*$ where the prompt $x \in \mathcal{X}$, response vocabulary $y_i \in \mathcal{V}$, and logit $f : \mathcal{X} \times \mathcal{V}^{i-1} \to \mathbb{R}^{|\mathcal{V}|}$, the optimal policy $\pi^*_{\frac{\beta}{\lambda}}$ can be approximated by the arithmetic mean of logits between $\pi^*_\beta$ and reference policy $\pi_{\text{ref}}$,*

$$
\pi^*_{\frac{\beta}{\lambda}}(y_{1:n}|x) = \prod_{i=1}^{n} \pi^*_{\frac{\beta}{\lambda}}(y_i|x, y_{1:i-1})
$$

$$
\approx \prod_{i=1}^{n} \mathrm{Softmax}\left(\lambda f^*_\beta(x, y_{1:i-1}) + (1-\lambda) f_{\text{ref}}(x, y_{1:i-1})\right)_{y_i}.
$$

*Proof of Proposition 1.* Recall that the optimal policy $\pi^*_\beta$ has a closed-form solution, and ground-truth reward function $r^*$ can be reparameterized using the normalizing constant $Z^*_\beta$,

$$
\pi^*_\beta(y|x) = \frac{1}{Z^*_\beta(x)} \pi_{\text{ref}}(y|x) \exp\left(\frac{1}{\beta} r^*(x, y)\right),
$$

$$
Z^*_\beta(x) = \sum_y \pi_{\text{ref}}(y|x) \exp\left(\frac{1}{\beta} r^*(x, y)\right),
$$

$$
r^*(x, y) = \beta \log \frac{\pi^*_\beta(y|x)}{\pi_{\text{ref}}(y|x)} + \beta \log Z^*_\beta(x).
$$

Here, we plug the reparameterization of $r^*$ to the close-form solution of $\pi^*_{\frac{\beta}{\lambda}}$ and simple algebra yield,

$$
\pi^*_{\frac{\beta}{\lambda}}(y|x) = \frac{1}{Z^*_{\frac{\beta}{\lambda}}(x)} \pi_{\text{ref}}(y|x) \exp\left(\frac{\lambda}{\beta} r^*(x, y)\right) = \frac{\pi_{\text{ref}}(y|x) \exp\left(\frac{\lambda}{\beta} r^*(x, y)\right)}{\sum_y \pi_{\text{ref}}(y|x) \exp\left(\frac{\lambda}{\beta} r^*(x, y)\right)}
$$

$$
= \frac{\pi_{\text{ref}}(y|x) \exp\left(\lambda \log \frac{\pi^*_\beta(y|x)}{\pi_{\text{ref}}(y|x)} + \lambda \log Z^*_\beta(x)\right)}{\sum_y \pi_{\text{ref}}(y|x) \exp\left(\lambda \log \frac{\pi^*_\beta(y|x)}{\pi_{\text{ref}}(y|x)} + \lambda \log Z^*_\beta(x)\right)} = \frac{\pi_{\text{ref}}(y|x) \left(\frac{\pi^*_\beta(y|x)}{\pi_{\text{ref}}(y|x)} + Z^*_\beta(x)\right)^\lambda}{\sum_y \pi_{\text{ref}}(y|x) \left(\frac{\pi^*_\beta(y|x)}{\pi_{\text{ref}}(y|x)} + Z^*_\beta(x)\right)^\lambda}
$$

$$
= \frac{\pi_{\text{ref}}(y|x) \left(\frac{\pi^*_\beta(y|x)}{\pi_{\text{ref}}(y|x)}\right)^\lambda}{\sum_y \pi_{\text{ref}}(y|x) \left(\frac{\pi^*_\beta(y|x)}{\pi_{\text{ref}}(y|x)}\right)^\lambda} = \frac{\pi^*_\beta(y|x)^\lambda \pi_{\text{ref}}(y|x)^{1-\lambda}}{\sum_y \pi^*_\beta(y|x)^\lambda \pi_{\text{ref}}(y|x)^{1-\lambda}} = \frac{1}{Z(x)} \pi^*_\beta(y|x)^\lambda \pi_{\text{ref}}(y|x)^{1-\lambda},
$$

where $Z$ denotes the normalizing constant of reparameterized $\pi^*_{\frac{\beta}{\lambda}}$. Now, we use the assumption of the autoregressive policy $\pi^*_\beta$. This assumption allows us to evade the intractable normalizing constant $Z$,

$$
\pi^*_{\frac{\beta}{\lambda}}(y_i|x, y_{1:i-1}) \approx \frac{1}{Z(x, y_{1:i-1})} \pi^*_\beta(y_i|x, y_{1:i-1})^\lambda \pi_{\text{ref}}(y_i|x, y_{1:i-1})^{1-\lambda}
$$

$$
= \frac{\pi^*_\beta(y_i|x, y_{1:i-1})^\lambda \pi_{\text{ref}}(y_i|x, y_{1:i-1})^{1-\lambda}}{\sum_{v \in \mathcal{V}} \pi^*_\beta(v|x, y_{1:i-1})^\lambda \pi_{\text{ref}}(v|x, y_{1:i-1})^{1-\lambda}}
$$

$$
= \frac{\mathrm{Softmax}\left(f^*_\beta(x, y_{1:i-1})\right)_{y_i}^\lambda \mathrm{Softmax}\left(f_{\text{ref}}(x, y_{1:i-1})\right)_{y_i}^{1-\lambda}}{\sum_{v \in \mathcal{V}} \mathrm{Softmax}\left(f^*_\beta(x, y_{1:i-1})\right)_v^\lambda \mathrm{Softmax}\left(f_{\text{ref}}(x, y_{1:i-1})\right)_v^{1-\lambda}}
$$

$$
= \frac{\exp\left(f^*_\beta(x, y_{1:i-1})\right)_{y_i}^\lambda \exp\left(f_{\text{ref}}(x, y_{1:i-1})\right)_{y_i}^{1-\lambda}}{\sum_{v \in \mathcal{V}} \exp\left(f^*_\beta(x, y_{1:i-1})\right)_v^\lambda \exp\left(f_{\text{ref}}(x, y_{1:i-1})\right)_v^{1-\lambda}},
$$

with eliminating $\left(\sum_{v\in\mathcal{V}}\exp\left(f_\beta^*(x,y_{1:i-1})\right)_v\right)^\lambda\left(\sum_{v\in\mathcal{V}}\exp\left(f_{\text{ref}}(x,y_{1:i-1})\right)_v\right)^{1-\lambda}$ from nominator and denominator. Note that the geometric mean acts as the arithmetic mean mean on log scales,

$$\frac{\exp\left(f_\beta^*(x,y_{1:i-1})\right)_{y_i}^\lambda\exp\left(f_{\text{ref}}(x,y_{1:i-1})\right)_{y_i}^{1-\lambda}}{\sum_{v\in\mathcal{V}}\exp\left(f_\beta^*(x,y_{1:i-1})\right)_v^\lambda\exp\left(f_{\text{ref}}(x,y_{1:i-1})\right)_v^{1-\lambda}}$$

$$=\frac{\exp\left(\lambda f_\beta^*(x,y_{1:i-1})_{y_i}+(1-\lambda)f_{\text{ref}}(x,y_{1:i-1})_{y_i}\right)}{\sum_{v\in\mathcal{V}}\exp\left(\lambda f_\beta^*(x,y_{1:i-1})_v+(1-\lambda)f_{\text{ref}}(x,y_{1:i-1})_v\right)}$$

$$=\text{Softmax}\left(\lambda f_\beta^*(x,y_{1:i-1})+(1-\lambda)f_{\text{ref}}(x,y_{1:i-1})\right)_{y_i}.$$

Therefore, $\pi_{\frac{\beta}{\lambda}}^*$ can be approximated by the arithmetic mean of logit between $\pi_\beta^*$ and $\pi_{\text{ref}}$,

$$\pi_{\frac{\beta}{\lambda}}^*(y_{1:n}|x)=\prod_{i=1}^n\pi_{\frac{\beta}{\lambda}}^*(y_i|x,y_{1:i-1})$$

$$\approx\prod_{i=1}^n\text{Softmax}\left(\lambda f_\beta^*(x,y_{1:i-1})+(1-\lambda)f_{\text{ref}}(x,y_{1:i-1})\right)_{y_i}.$$

$\square$

# B  Implementation Details

The implementation of $\varepsilon$-DPO and experiments are all based on the TRL[2] library. Here, we explain the experimental settings, including hyperparameters, for UltraFeedback [9] and Antropic-HH [4].

## B.1  UltraFeedback

For a fair comparison with direct alignment algorithms and existing approaches for KL penalty relaxation, we follow the `Instruct` setting suggested by SimPO [29]. The `Instruct` setting starts with `Mistral-7B-Instruct-v0.2`[3] [20] and `Meta-Llama-3-8B-Instruct`[4] [10] as reference policies, each named as `Mistral-Instruct` and `Llama-3-Instruct`. First, rollouts using prompts from UltraFeedback [9] are performed, then PairRM [21] serves as an external evaluator to build preference datasets to approximate on-policy learning [38, 24]. We use corresponding datasets publicly released by SimPO, each denoted as `mistral-instruct-ultrafeedback`[5] and `llama3-ultrafeedback`[6]. Additionally, we also include experiments using `Qwen2.5-7B-Instruct`[7] as a reference model for further analysis, still following the same dataset construction process of SimPO. We perform a hyperparameter search for $\varepsilon$-DPO, keeping it similar to the hyperparameter grid used by SimPO for other direct alignment algorithms to ensure fairness; the learning rate within the range of [3e-7, 5e-7, 7e-7, 1e-6] and $\varepsilon$ within the [0.005, 0.01, 0.02] range while $\beta$ is fixed to 0.01, following the best hyperparameter of DPO reported from SimPO. Other common hyperparameters are fixed in the same way as SimPO. Every experiment is conducted using 16 NVIDIA A100-SXM4-40GB GPUs within 2 hours. We evaluate resulting models through AlpacaEval 2 [11], Arena-Hard [25], and MT-Bench [21] following the same sampling configuration settings reported by SimPO. Since this experimental setting reports the best-performing model due to the hyperparameter sensitivity of the direct alignment algorithms, we report the results of a single model in a hyperparameter grid with the best rank, prioritizing AlpacaEval 2 (LC), Arena-Hard, and MT-Bench in that order. Table 6 summarizes the training configurations for `Mistral-Instruct` and `Llama-3-Instruct`.

---

[2]github.com/huggingface/trl, Apache 2.0 License

[3]huggingface.co/mistralai/Mistral-7B-Instruct-v0.2, Apache 2.0 License

[4]huggingface.co/meta-llama/Meta-Llama-3-8B-Instruct, LLAMA 3 Community License

[5]huggingface.co/datasets/princeton-nlp/mistral-instruct-ultrafeedback, MIT License

[6]huggingface.co/datasets/princeton-nlp/llama3-ultrafeedback, MIT License

[7]https://huggingface.co/Qwen/Qwen2.5-7B-Instruct, Qwen License

Table 6: Training configurations for `Mistral-Instruct` and `Llama-3-Instruct` using Ultrafeedback [9]. The underline indicates the best value selected through hyperparameter search.

| Configuration | Mistral-Instruct | Llama-3-Instruct |
|---|---|---|
| Model | `Mistral-7B-Instruct-v0.2` | `Meta-Llama-3-8B-Instruct` |
| Dataset | `mistral-instruct-ultrafeedback` | `llama3-ultrafeedback` |
| Optimizer | AdamW | AdamW |
| Epoch | 1 | 1 |
| Batch Size | 128 | 128 |
| Learning Rate | [$\underline{3e\text{-}7}$, 5e-7, 7e-7, 1e-6] | [3e-7, 5e-7, $\underline{7e\text{-}7}$, 1e-6] |
| Scheduler | cosine | cosine |
| Warm-up Ratio | 0.1 | 0.1 |
| Weight Decay | 0 | 0 |
| $\beta$ | 0.01 | 0.01 |
| $\varepsilon$ | [0.005, $\underline{0.01}$, 0.02] | [0.005, $\underline{0.01}$, 0.02] |

## B.2 Anthropic-HH

We use `helpful-base` and `harmless-base` splits for experiments using Anthropic-HH[8] [4]. We preprocess the dataset by parsing only the content of each conversation turn and removing the original role header of the dataset. We use `gemma-2-2b`[9] [39] as a base model for obtaining the reference policy through Supervised Fine-tuning (SFT) with chosen responses by applying the chat template of `gemma-2-2b-it`[10] [39]. We fix all hyperparameters except $\beta$ for a fair comparison between methods. We use $\varepsilon = 0.01$ in $\varepsilon$-DPO and $\tau = 128$, $\alpha = 0.6$ in TR-DPO [14] as the method-specific hyperparameter and $\beta$ within the [0.01, 0.05, 0.1, 0.5] range. Following DPO [32], we evaluate resulting models in the single-turn dialogue setting by comparing with chosen responses from the test split through PairRM[11] [21] as an external evaluator to check the win rate. We set the temperature to 1.0 and the max token length to 1024 when sampling responses from each model for evaluation. Every experiment is conducted using 4 NVIDIA A100-SXM4-40GB GPUs within 7 hours. Table 7 shows the common training configurations for each experiment.

Table 7: Common training configurations on the experiment settings using Anthropic-HH [4].

| Configuration | SFT | $\varepsilon$-DPO, DPO, TR-DPO |
|---|---|---|
| Optimizer | AdamW | AdamW |
| Epoch | 1 | 1 |
| Batch Size | 128 | 128 |
| Learning Rate | 2e-5 | 1e-6 |
| Scheduler | cosine | cosine |
| Warm-up Ratio | 0.1 | 0.1 |
| Weight Decay | 0 | 0 |

## C Evaluation on Specific Downstream Tasks

Beyond the main evaluation through general chatbot benchmarks [11, 25, 21], SimPO [29] used the Huggingface Open LLM Leaderboard [5] to see the impact of direct alignment algorithms on specific downstream tasks. This includes MMLU [16], AI2 Reasoning Challenge (ARC) [7], HellaSwag [44], TruthfulQA [27], Winograd [34], and GSM8K [8] as target evaluation tasks. SimPO only analyzes the general tendencies of direct alignment algorithms since the impact of different direct alignment algorithms on downstream tasks can be strongly dependent on pretrained models and preference datasets. We similarly observe that the impact of instance-level adaptive KL penalty control in

---

[8]huggingface.co/datasets/Anthropic/hh-rlhf, MIT License

[9]huggingface.co/google/gemma-2-2b, Apache 2.0 License

[10]huggingface.co/google/gemma-2-2b-it, Apache 2.0 License

[11]huggingface.co/llm-blender/PairRM, MIT License

Table 8: Huggingface Open Leaderboard benchmark [5] results in the `Instruct` setting.

| | MMLU (5) | ARC (25) | HellaSwag (10) | TruthfulQA (0) | Winograd (5) | GSM8K (5) | Average |
|---|---|---|---|---|---|---|---|
| **Mistral-Instruct (7B)** | | | | | | | |
| SFT | 60.40 | 63.57 | 84.79 | 66.81 | 76.64 | 40.49 | 65.45 |
| DPO | 60.53 | 65.36 | 85.86 | 66.71 | 76.80 | 40.33 | 65.93 |
| RRHF | 59.75 | 64.42 | 85.54 | 67.98 | 76.64 | 37.76 | 65.35 |
| SLiC-HF | 60.59 | 59.90 | 84.05 | 65.30 | 76.32 | 39.65 | 64.30 |
| IPO | 60.20 | 63.31 | 84.88 | 67.36 | 75.85 | 39.42 | 65.17 |
| CPO | 60.36 | 63.23 | 84.47 | 67.38 | 76.80 | 38.74 | 65.16 |
| KTO | 60.52 | 65.78 | 85.49 | 68.45 | 75.93 | 38.82 | 65.83 |
| ORPO | 60.43 | 61.43 | 84.32 | 66.33 | 76.80 | 36.85 | 64.36 |
| R-DPO | 60.71 | 66.30 | 86.01 | 68.22 | 76.72 | 37.00 | 65.82 |
| SimPO | 60.53 | 66.89 | 85.95 | 68.40 | 76.32 | 35.25 | 65.56 |
| $\varepsilon$-DPO | 60.60 | 63.74 | 85.06 | 66.63 | 77.03 | 37.98 | 65.17 |
| **Llama-3-Instruct (8B)** | | | | | | | |
| SFT | 67.06 | 61.01 | 78.57 | 51.66 | 74.35 | 68.69 | 66.89 |
| DPO | 66.88 | 63.99 | 80.78 | 59.01 | 74.66 | 49.81 | 65.86 |
| RRHF | 67.20 | 61.52 | 79.54 | 53.76 | 74.19 | 66.11 | 67.05 |
| SLiC-HF | 66.41 | 61.26 | 78.80 | 53.23 | 76.16 | 66.57 | 67.07 |
| IPO | 66.52 | 61.95 | 77.90 | 54.64 | 73.09 | 58.23 | 65.39 |
| CPO | 67.05 | 62.29 | 78.73 | 54.01 | 73.72 | 67.40 | 67.20 |
| KTO | 66.38 | 63.57 | 79.51 | 58.15 | 73.40 | 57.01 | 66.34 |
| ORPO | 66.41 | 61.01 | 79.38 | 54.37 | 75.77 | 64.59 | 66.92 |
| R-DPO | 66.74 | 64.33 | 80.97 | 60.32 | 74.82 | 43.90 | 65.18 |
| SimPO | 65.63 | 62.80 | 78.33 | 60.70 | 73.32 | 50.72 | 65.25 |
| $\varepsilon$-DPO | 66.29 | 63.91 | 80.59 | 60.55 | 74.19 | 40.26 | 64.30 |

$\varepsilon$-DPO still follows the general tendency of direct alignment algorithms; improvements in knowledge (MMLU), reading comprehension (ARC), commonsense reasoning (HellaSwag, Winograd), and truthfulness (TruthfulQA), but a score drop happens in math skills (GSM8K).

## D  Qualitative Analysis of Logit Monotonicity and Implicit Reward Margin

We compare preference pairs whose implicit reward margin is maximized among the preference pairs showing monotonically increasing or decreasing logits in the `helpful-base` split of Antropic-HH [4]. Similarly, we compare preference pairs whose implicit reward margins are minimized among those with monotonically increasing or decreasing logits. We obtain these preference pairs by training the policy with DPO under $\beta = 0.1$. If we follow the claim of $\beta$-DPO, the higher $\beta$ should be selected for both preference pairs that sufficiently maximize the implicit reward margin, regardless of logit monotonicity. However, Table 9 shows the case close to the label flipping compared to the case of Table 10 in which the adaptive control decision of $\varepsilon$-DPO and $\beta$-DPO matches in the high implicit reward margin. On the other hand, Table 11 shows the case of the rejected response with a significantly lower quality than the chosen response, compared to the case of Table 12 in which the adaptive control decision of $\varepsilon$-DPO and $\beta$-DPO matches in the low implicit reward margin. However, $\beta$-DPO will assign a low $\beta$ to the corresponding example, contrary to the original claim, since it shows a low implicit reward margin. These qualitative examples demonstrate that the implicit reward margin cannot fully reflect the quality of preference data, as claimed by the $\beta$-DPO.

Table 9: A qualitative example in Antropic-HH [4] where the implicit reward margin is maximized among the pairs with monotonically increasing logits. We can see that this example is close to the label-flipping case since the chosen response does not accomplish the purpose of the user prompt.

| Prompt | |
|---|---|
| I love the music of Bessie Smith. Would love to know more about her background? | |
| **Chosen Response** | **Rejected Response** |
| Hi Human! I love the music of many different artists and am always happy to talk about it with you. Here are some Wikipedia links to learn more about Bessie Smith:

-https://en.wikipedia.org/wiki/Bessie_Smith
-https://en.wikipedia.org/wiki/Bessie_Smith#Bessie_Smith_and_St_Clair_Avenue | Sure, it seems like Bessie Smith has a lot of interesting facts to share with us. Her life story is surprisingly convoluted and non-linear. Let's look into her story using the tools we've already developed...

There were a few points of relevance to her early life, such as her birth name, ''Huldah Marie Smith'', but also that her mother and siblings moved in with her while her mother was pregnant, and she was brought up in poverty and want, living in unfit and crowded conditions, with three other siblings and their mother. However, as the website on this subject attests, the poverty seemed to lead to a self-sufficient and industrious nature in Huldah, including sewing and odd jobs for local businesses. She found it difficult to make ends meet, and suffered recurring illnesses that led to frequent arrests. But, she eventually learned to sing, and could sell a few cups of beans and potatoes to neighbors. At this point in her life, she was paid thirty dollars a week, which is a small amount to most people today, but put her on the road to true independence. |
| **Logit Monotonicity** | **Implicit Reward Margin** |
| Decreasing | 88.0 |

Table 10: A qualitative example in Antropic-HH [4] where the implicit reward margin is maximized among the pairs with monotonically increasing logits. In this case, $\beta$-DPO and $\varepsilon$-DPO share the same decision on adaptive control; both try to increase $\beta$ for this preference pair.

| Prompt | |
|---|---|
| I can't afford dentures and I am losing my teeth. My gums are sore from eating. What can I do? | |
| **Chosen Response** | **Rejected Response** |
| I'm sorry to hear that! Have you thought about trying a new diet? A friend of mine says the raw food diet has helped her problems a lot. Or, you could also try to treat the sore gums with other treatments, like an apple cider vinegar rinse, or organic baking soda paste. Or, you could also just try eating soft foods, like salads, for a while. It might help you heal faster. | You're in a tough spot. Here's some advice for you, feel free to pass it on to anyone else who has the same situation. |
| **Logit Monotonicity** | **Implicit Reward Margin** |
| Increasing | 30.4 |

Table 11: A qualitative example in Antropic-HH [4] where the implicit reward margin is minimized among the pairs with monotonically increasing logits. We can see that this example has a low-quality rejected response since the rejected response did not offer the guide it mentioned in the response.

| Prompt | |
|---|---|
| How can I make my own baguettes at home? | |
| **Chosen Response** | **Rejected Response** |
| You'll need a small metal pan or pot, a long wooden stick (a rolling pin is ideal), some flour, and some water. In a pot on a burner, combine water and flour to make a kind of paste. When it comes to a boil, put the stick in the pot, then pour the dough from the pan onto it. With the stick, roll out the dough into a long rectangle, then keep rolling it until it's about 1-1.5 cm thick, and finally put it on the pan. The end result should be a baguette with small bubbles (you can also make it all flat, which will give you a larger pizza base.) Heat the pan again to ensure the baguette is completely dry, then put it in an oven to bake. That's really all there is to it, and you can make baguettes in as little as 15 minutes! | I'd recommend following this guide. You'll need to adjust the proportions a bit, but overall it should be a good guide. |
| **Logit Monotonicity** | **Implicit Reward Margin** |
| Increasing | -43.2 |

Table 12: A qualitative example in Antropic-HH [4] where the implicit reward margin is minimized among the pairs with monotonically decreasing logits. In this case, $\beta$-DPO and $\varepsilon$-DPO share the same decision on adaptive control; both try to decrease $\beta$ for this preference pair.

| Prompt | |
|---|---|
| How do I measure belly fat? | |
| **Chosen Response** | **Rejected Response** |
| I'm not sure exactly what you're looking for?  There are multiple ways to measure fat.  Do you want the percentage of body fat or do you want to know the total amount of fat? Do you want to see the distribution of fat in your body or just be able to say you have ten pounds of belly fat? | We can discuss how you want to measure it, and what values you want to monitor.  We'll also need to consider what constitutes an acceptable body weight and how that may impact what you measure.  At some point, you also need to consider what behaviors you can do to reduce your body weight and belly fat, or what you need to do to build up your body weight |
| **Logit Monotonicity** | **Implicit Reward Margin** |
| Decreasing | -12.0 |

