# OpenReview forum: "KL Penalty Control via Perturbation for Direct Preference Optimization"
_NeurIPS.cc/2025/Conference — NeurIPS 2025 poster_

### Official Review · Reviewer_4tbE · 2025-06-23

**Clarity:** 3
**Significance:** 2
**Originality:** 2
**Rating:** 4
**Confidence:** 3

**Summary:**

This paper proposes a variant of DPO, termed $\varepsilon$-DPO, which introduces an adaptive mechanism to control the KL penalty parameter $\beta$. By perturbing the KL term at the instance level, $\varepsilon$-DPO dynamically balances the trade-off between staying close to the reference model and aligning with preference data. Experimental results demonstrate that $\varepsilon$-DPO outperforms existing baselines, achieving improved alignment and generalization in preference optimization tasks.

**Questions:**

1. Do equations (1), (2), and (5) imply that the choice of $\beta_ \varepsilon^+$ and $\beta_ \varepsilon^-$ are determined by the sign of  $z_\theta(x,y^w,y^l)$?
2. For Equation (4), I understand that $\pi_{\theta(\beta_ \varepsilon^+)}$ can be interpreted as the softmax over a convex combination of the logits $f_\theta$ and $f_{\textnormal{ref}}$. However, for Equation (3), the interpretation is less clear. Could the authors provide an intuitive explanation of this expression?

**Ethical Concerns:**

["NO or VERY MINOR ethics concerns only"]

**Final Justification:**

The response has addressed all my concerns, and I raised my score for acceptance.

**Limitations:**

The authors have discussed the limitations in conclusion and I raise my score for acceptance.

**Quality:**

2

**Strengths And Weaknesses:**

Strengths:

The paper introduces a principled and effective way to adaptively control the KL penalty parameter $\beta$ in DPO through instance-level perturbations. This addresses a key limitation of standard DPO, which relies on a fixed penalty term, and allows the model to better accommodate diverse preference samples.

Weaknesses:
1. While the introduction of an adaptive KL penalty in DPO is a useful refinement, the core idea, which modulates the regularization strength based on instance-level signals, may be seen as a relatively incremental extension of existing approaches. The paper would benefit from a stronger emphasis on the broader implications or theoretical insights enabled by this adaptation.
2. The theoretical development builds heavily on existing results and offers limited additional insight into the underlying mechanisms that make the proposed method effective. A more principled analysis of why and when the approach works would strengthen the contribution. (See details in the question part)

---

> ### Author Rebuttal · Authors · 2025-07-31
>
> **1. Clarification of Theoretical Insights Offered from $\varepsilon$-DPO [W1]**
>
> > **[W1]** While the introduction of an adaptive KL penalty in DPO is a useful refinement, the core idea, which modulates the regularization strength based on instance-level signals, may be seen as a relatively incremental extension of existing approaches. The paper would benefit from a stronger emphasis on the broader implications or theoretical insights enabled by this adaptation.
>
> We sincerely appreciate your feedback on how we can enhance the messages enabled by our work. Proposition 1, which we use as a key property in $\varepsilon$-DPO, is motivated by the observation of [1], which is influenced by [2]. **Still, both studies aim to approximate policies that have not been trained at sampling time, but our work offers a novel theoretical insight in the different aspect; this sampling-time approximation approach can be effectively utilized at training-time for estimating an appropriate KL penalty $\beta$, by approaching it from the perspective of preference model reparameterization of the DPO.** While [1] aims to control the KL penalty coefficient $\beta$ used in DPO at test-time and [2] estimates the importance ratio for instruction-following of a large-scale model through an implicit reward defined between a small-scale instruction-tuned model and a pretrained model, both approaches could not reach the theoretical insights that the current policy model, which has not completely ended the training process can still perfomrs approximation of other policy for controlling regularization during training process. **Simultaneously, it achieves instance-level adaptive KL penalty control, which is not possible with existing approaches like $\beta$-DPO, which determines the KL penalty control criterion through empirical observation, or TR-DPO, which approaches it based on the property in the loss landscape.** We will reflect that this new theoretical insight is more clearly reflected in the final version, as suggested.
>
> **2. Detailed Interpretation of Theoretical Insights of $\varepsilon$-DPO [W2, Q1, Q2]**
>
> > **[W2]** The theoretical development builds heavily on existing results and offers limited additional insight into the underlying mechanisms that make the proposed method effective. A more principled analysis of why and when the approach works would strengthen the contribution. (See details in the question part)
>
> > **[Q1]** Do equations (1), (2), and (5) imply that the choice of $\beta_{\varepsilon}^{+}$ and $\beta_{\varepsilon}^{-}$ are determined by the sign of $z_\theta(x, y^w, y^l)$?
>
> > **[Q2]** For Equation (4), I understand that $\pi_{\theta(\beta_{\varepsilon}^{+})}$
>  can be interpreted as the softmax over a convex combination of the logits $f_\theta$
>  and $f_\text{ref}$. However, for Equation (3), the interpretation is less clear. Could the authors provide an intuitive explanation of this expression?
>
> We sincerely appreciate your effort to find the theoretical insights of $\varepsilon$-DPO. First, we can intuitively understand checking (1) and (2) for a given triplet $(x, y^w, y^l)$ in the two perspective: i) the direction of the training-time inverse temperature $\beta$ that induces better separability of preference model defined by the optimal policy $\pi_{\theta(\beta)}$, and ii) the sign of the numerical derivative with respect to $\beta$ that maximizes the intractable logit function $z(x, y^w, y^l; \theta(\beta))$, as described in Line 129 - 135. **However, (1) and (2) cannot be verified simply through the sign of the logit from the current policy $z_\theta(x, y^w, y^l)$, but through the access to the optimal policy $\pi_{\theta(\beta)}$ for all possible $\beta$, regardless of the interpretations of i) and ii).** In other words, (1) and (2) can be verified if we have all the policies that have already been trained for all $\beta$. In this context, we utilize Proposition 1 as (3) and (4) to approximately verify that (1) and (2) are satisfied even though we do not have policies that have been trained for all possible $\beta$. By comparing the magnitude of the logit of the optimal policy estimated under $\beta_{\varepsilon}^{+}$ and $\beta_{\varepsilon}^{-}$ near the current $\beta$ according to Proposition 1 with the logit estimated at the current $\beta$, $\beta_{\varepsilon}^{+}$ and $\beta_{\varepsilon}^{-}$ are determined. **That is, $\beta_{\varepsilon}^{+}$ and $\beta_{\varepsilon}^{-}$ are not determined by the sign of the logit from current policy  $z_\theta(x, y^w, y^l)$, but through the comparison between estimated logits of the policies under perturbation of $\beta$**. Meanwhile, if we interpret (3) and (4) in a context of 'convex combination' as you mentioned for (3), they can be understood as obtaining points inside the convex hull (i.e., (3)) or outside the convex hull (i.e., (4)) on the affine hull defined in logit space. **If we consider points inside the convex hull as a kind of 'interpolation' in the logit space, then, unlike (3), (4) can be understood as an 'extrapolation' in the logit space.** However, since Proposition 1 is derived from the following relationship of optimal policies for importance sampling, we believe that interpreting (3) and (4) as perturbations of the modulation parameter $\lambda$ for the importance ratio is somewhat more in line with the original theoretical intention.
>
> $ \pi_{\theta(\frac{\beta}{\lambda})}(y_i|x, y_{1:i-1}) \propto \pi_{\text{ref}}(y_i|x, y_{1:i-1}) [\frac{\pi_{\theta(\beta)}(y_i|x, y_{1:i-1})}{\pi_{\text{ref}}(y_i|x, y_{1:i-1})}]^\lambda $
>
> _____
>
> [1] Decoding-time Realignment of Language Models (Liu et al., 2024)
> [2] An Emulator for Fine-Tuning Large Language Models using Small Language Models (Mitchell et al., 2023)

---

> > ### Comment · Reviewer_4tbE · 2025-08-04
> >
> > Thank you for your detailed response. It resolved all my concerns, so I will raise my score to 4.

---

### Official Review · Reviewer_gyoi · 2025-06-30

**Clarity:** 3
**Significance:** 2
**Originality:** 3
**Rating:** 4
**Confidence:** 4

**Summary:**

This paper introduces ϵ-Direct Preference Optimization (ϵ-DPO), a new method for aligning large language models with human preferences. The key innovation of ϵ-DPO is its ability to adaptively control the strength of the KL penalty for each individual data point (instance-level) during training. This contrasts with standard Direct Preference Optimization (DPO), which uses a fixed KL penalty, and other methods that adjust it at a batch-level or through periodic, non-adaptive updates.

The core mechanism of ϵ-DPO involves checking the "monotonicity of logits." Essentially, for each preference pair (a chosen and a rejected response), the method observes how the model's confidence (represented by the log-likelihood ratio) changes when the KL penalty coefficient, β, is slightly perturbed. This check allows the algorithm to decide whether to increase or decrease the KL penalty for that specific training example, without requiring extra model updates or being dependent on the batch size. The policy under this perturbed penalty is efficiently estimated by reusing the logits from the current and reference policies.

**Questions:**

1. In Line 232, what is step size $\mathbb{E}(\epsilon_i)$? Where is $\epsilon_i$ defined? How is this related to $\epsilon_c$ and $\epsilon_s$?
2. Can you analyze why on the same dataset, e.g. Arena-Hard, Mistral and Llama will have very different results? (in Table 1), could you provide experiments on more models? such as Qwen?
3. Does computing (3) (4) requires more computational resources compared with the DPO objective? In other words, how much more FlOPs are needed (in percentile) in addition to the DPO objective?
4. In "Variants of KL Penalty Control Strategy", the authors claim that we should set $\epsilon_c=\epsilon_\beta$. In the Table 3, it sets $\epsilon_c=\epsilon_\beta=0.01$. Can you set the fixed $\epsilon$ to other values? such as when $\epsilon_c$ or $\epsilon_\beta$ equals $0.005, 0.01,0.02$, and varying another one. Or in general, you should enumerate all choices of $\epsilon_c$, $\epsilon_\beta$, in $0.005,0.01, 0.02$ (or more) and say $\epsilon_c=\epsilon_\beta=0.01$ is the best choice.

I am willing to increase my score if the weaknesses and questions are properly resolved.

**Ethical Concerns:**

["NO or VERY MINOR ethics concerns only"]

**Final Justification:**

I increase my score because the rebuttal solves my problem. However, since $\epsilon$-DPO's inferior performance on math reasoning datasets compared with DPO. I decide to increase my score to 4 instead of 5.

**Limitations:**

Yes.

**Paper Formatting Concerns:**

There are no major formatting issues.

**Quality:**

3

**Strengths And Weaknesses:**

### Strength
- The experimental results show that ϵ-DPO doesn't only improve upon DPO but also outperforms most existing direct alignment algorithms on several key chatbot benchmarks. The paper demonstrates that ϵ-DPO achieves a more efficient Pareto frontier between performance and KL divergence compared to DPO and TR-DPO. This shows that the adaptive control is not just indiscriminately increasing KL divergence but is doing so efficiently for performance gains.
- The paper provides strong evidence that the static KL penalty in DPO is a "major bottleneck to final model performance". The success of ϵ-DPO underscores the importance of instance-level adaptive control for unlocking better performance.
- While the idea of a dynamic KL penalty is not new , this paper is the first to propose a method that is both instance-level and adaptive.

### Weakness
 - The method's effectiveness hinges on the quality of the policy approximation in Proposition 1. The authors acknowledge that a large perturbation size $\epsilon$ can lead to a "weaker approximation" and potentially "a wrong decision for KL penalty relaxation". The paper could be strengthened by a more in-depth analysis of the error bounds or the conditions under which this approximation is most reliable.
- As the authors correctly state in their limitations section, ϵ-DPO, being an extension of DPO, requires a reference policy. This entails additional memory consumption and computation compared to reference-free methods like ORPO and SimPO.
- The evaluation on specific downstream tasks in Appendix C shows that ϵ-DPO has a performance drop compared with DPO on math reasoning (GSM8K). The average scores reported in Table 6 do not show a consistent, decisive advantage for ϵ-DPO, suggesting its benefits are most pronounced in conversational ability rather than across all specialized tasks.
- $\beta$ for the next training step is updated by taking the mean of the instance-level beta values used in the current batch. I think this could be replaced by a potentially more robust update scheme (e.g., an exponential moving average with a learnable parameter).

---

> ### Author Rebuttal · Authors · 2025-07-31
>
> **1. Conditions for Reliable Approximation [W1]**
>
> > **[W1]** The method's effectiveness hinges on the quality of the policy approximation in Proposition 1. The authors acknowledge that a large perturbation size $\epsilon$ can lead to a "weaker approximation" and potentially "a wrong decision for KL penalty relaxation". The paper could be strengthened by a more in-depth analysis of the error bounds or the conditions under which this approximation is most reliable.
>
> We sincerely appreciate your suggestion to include an in-depth analysis of the adaptive control criterion. To verify how much the robust estimation can be satisfied during the training process, we additionally checked the average upper bound of $\varepsilon$ that define the neighborhood $(-\varepsilon, +\varepsilon)$ that the all value in the neighborhood consistently satisfies the logit monotonic criterion for triplets $(x, y^w, y^l)$ of test splits in Antropich-HH, through checkpoints obtained in 0.1 epoch intervals on DPO under $\beta = 0.05$ and `gemma-2-2b`. **This corresponds to checking the $\varepsilon$ bound of consistency on logit monotonicity according to approximation strengths**. We tested 100 uniform sample points of $\varepsilon$ within the range (0.005, 0.02). **We observed that the expected upper bound of $\varepsilon$, $\varepsilon_{\downarrow}$ and $\varepsilon_{\uparrow}$, which correspond to the monotonic decreasing and increasing logits, respectively, converged almost at 0.008 after 0.2 epochs, similar to the best result of $\varepsilon=0.01$ of previous experiments, as below.** Furthermore, the lower value of $\varepsilon_{\downarrow}$ and $\varepsilon_{\uparrow}$ at 0.1 epoch is also consistent with the degeneration phenomenon discussed in Section 4.3. **Therefore, we can confirm that the logit monotonicity criterion achieves reliable estimations during training steps with an appropriate $\varepsilon$ level, excluding the early training steps.**
>
> | Upper Bound of $\varepsilon$ | 0.1 | 0.2 | 0.3 | 0.4 | 0.5 | 0.6 | 0.7 | 0.8 | 0.9 | 1.0|
> |-|:-:|:-:|:-:|:-:|:-:|:-:|:-:|:-:|:-:|:-:|
> | $\varepsilon_{\downarrow}$ (Avg / Std) | 0.0068 / 0.0011 | 0.0080 / 0.0038 | 0.0081 / 0.0040 | 0.0080 / 0.0040 | 0.0080 / 0.0040 | 0.0084 / 0.0044 | 0.0087 / 0.0047 | 0.0083 / 0.0041 | 0.0083 / 0.0041 | 0.0081 / 0.0038 |
> | $\varepsilon_{\uparrow}$ (Avg / Std) | 0.0067 / 0.0006 | 0.0076 / 0.0032 | 0.0079 / 0.0037 | 0.0079 / 0.0037 | 0.0079 / 0.0038 | 0.0079 / 0.0038 | 0.0084 / 0.0043 | 0.0079 / 0.0043 | 0.0082 / 0.0037 | 0.0083 / 0.0040 |
>
> **2. Analysis of Additional Computation Cost [W2, Q3]**
>
> > **[W2]** As the authors correctly state in their limitations section, ϵ-DPO, being an extension of DPO, requires a reference policy. This entails additional memory consumption and computation compared to reference-free methods like ORPO and SimPO.
>
> > **[Q3]** Does computing (3) (4) requires more computational resources compared with the DPO objective? In other words, how much more FlOPs are needed (in percentile) in addition to the DPO objective?
>
> We sincerely appreciate your careful observation of the additional computational cost. **Despite $\varepsilon$-DPO requiring reference models, it can save computation through pre-computing the logits of the reference model, similar to DPO.** Also, $\varepsilon$-DPO requires additional computations of (3) and (4) compared to DPO, each of which involves two scalar-vector multiplications, vector additions, and log-softmax operations per token; a total $16v$ FLOPs are added per token. **On the other hand, FLOPs per token of DPO, which is approximately $8N$ for a given model parameter size $N$, excluding the embedding layer [1], thus resulting in $\frac{200v}{N} (\\%)$ additional FLOPs per token, which is negligible since generally $v << N$ holds.** We also compared the wall-time increment $\Delta t$ of $\varepsilon$-DPO compared to DPO in `Mistral-Instruct` and `Llama-Instruct` settings to provide an intuitive comparison as below. We can find that $\varepsilon$-DPO results in a negligible wall-time increment.
>
> | Wall-time Increment | Mistral-Instruct | Llama-Instruct |
> |-|:-:|:-:|
> | Step (Avg) | 0.00082s | 0.00064s |
> | Epoch | 0.38076s | 0.30016s |
> | $\frac{\Delta t}{t} (\\%) $ | 0.0064% | 0.0045% |
>
> **3. Performance Differences of $\varepsilon$-DPO across Base Models [W3, Q2]**
>
> > **[W3]** The evaluation on specific downstream tasks in Appendix C shows that ϵ-DPO has a performance drop compared with DPO on math reasoning (GSM8K). The average scores reported in Table 6 do not show a consistent, decisive advantage for ϵ-DPO, suggesting its benefits are most pronounced in conversational ability rather than across all specialized tasks.
>
> > **[Q2]** Can you analyze why on the same dataset, e.g. Arena-Hard, Mistral and Llama will have very different results? (in Table 1), could you provide experiments on more models? such as Qwen?
>
> We sincerely appreciate your careful comparisons on the performance of $\varepsilon$-DPO. **We want to emphasize that the experimental settings in UltraFeedback focus on alignment as conversational agents, rather than specific downstream tasks at the dataset level.** We speculate that if $\varepsilon$-DPO is adopted to target the mathematical reasoning dataset [2], it can sufficiently show results different from our experiments. **Furthermore,  trends on specific benchmarks are likely to be influenced by the characteristics of the base model, because different datasets are constructed by sampling responses from each model [3].** Due to the limited computation resources, we fix the best method-specific hyperparameters of DPO, SimPO, and $\varepsilon$-DPO obtained in the `Llama-Instruct` setting and trained `Qwen2.5-7B-Instruct` with a hyperparameter search range of learning rate in [3e-7, 5e-7, 7e-7] and the same annotation process as suggested. The results below show that $\varepsilon$-DPO achieves concrete performance as in the `Llama-Instruct` setting.
>
> | Method |AlpacaEval 2 (LC / WR) | Arena-Hard (WR) | MT-Bench (Score) |
> |-|:-:|:-:|:-:|
> | SFT | 27.8 / 27.9 | 51.8 | 8.6 |
> | DPO | 41.6 / **46.3** | 66.8 | 8.9 |
> | SimPO | 32.4 / 46.0 | 60.2 | 8.8 |
> | $\varepsilon$-DPO | **42.5** / 46.1 | **67.5** | **9.1** |
>
> **4. Alternative for the Updating Rule of $\beta$ [W4]**
>
> > **[W4]** $\beta$ for the next training step is updated by taking the mean of the instance-level beta values used in the current batch. I think this could be replaced by a potentially more robust update scheme (e.g., an exponential moving average with a learnable parameter).
>
> We sincerely appreciate your valuable suggestions for improving $\varepsilon$-DPO. We also agree that EMA with a learnable parameter can be a powerful alternative. **However, it introduces a new hyperparameter, $m$, for the smoothing factor. Furthermore, we experimentally confirmed that letting $\beta$ be learnable without a $\beta$-specific learning rate resulted in almost no variation of $\beta$, because of the relatively low learning rate of the direct alignment algorithm, resulting in another difficulty of training.** Still, EMA allows moderate $\beta$ variation, so we experimented with the EMA updating of $\beta$ using $m = 0.5$, in addition to the best hyperparameters obtained from `Mistral-Instruct` and `Llama-Instruct`. Unfortunately, we observed degeneration compared to the result of $\varepsilon$-DPO, as shown below.
>
> | Setting | Method |AlpacaEval 2 (LC / WR) | Arena-Hard (WR) | MT-Bench (Score) |
> |-|-|:-:|:-:|:-:|
> | Mistral-Instruct | $\varepsilon$-DPO (EMA) | 23.1 / 18.8 | 15.8 | 7.6 |
> | Mistral-Instruct | $\varepsilon$-DPO| **35.6** / **29.6** | **17.2** | **7.8** |
> | Llama-Instruct | $\varepsilon$-DPO (EMA) | 38.6 / 36.7 | 35.6 | **8.0** |
> | Llama-Instruct | $\varepsilon$-DPO| **42.5** / **44.9** | **36.7** | **8.0** |
>
>
> **5. Clarification of the Notation and Experiments [Q1, Q4]**
>
> > **[Q1]** In Line 232, what is step size $\mathbb{E}(\epsilon_i)$? Where is $\epsilon_i$ defined? How is this related to $\epsilon_c$ and $\epsilon_s$?
>
> > **[Q4]** In "Variants of KL Penalty Control Strategy", the authors claim that we should set
> . In the Table 3, it sets $\epsilon_c = \epsilon_s$. Can you set the fixed
>  to other values? such as when $\epsilon_c$ or $\epsilon_s$ equals $0.005, 0.01, 0.02$, and varying another one. Or in general, you should enumerate all choices of $\epsilon_c$, $\epsilon_s$ in $0.005, 0.01, 0.02$ (or more) and say $\epsilon_c = \epsilon_s = 0.01$ is the best choice.
>
> We sincerely appreciate your comments on the clarification about the notation. We found that the existing notation was insufficiently illustrative and potentially misleading. **$\mathbb{E}[\varepsilon_i]$ represents the average of occurrences of $\beta^+$ and $\beta^-$ by regarding each as +1 and -1, respectively, and is intended to illustrate the overall training dynamics according to the choice of $\varepsilon_c$ and $\varepsilon_s$.** We will reflect the corrections and explanations of the notation in the final version. Furthermore, we also experimented to compare all choices in $\varepsilon=[0.005, 0.01, 0.02]$ to provide better ablation, as you suggested. In this process, we found an error in the evaluation at the case of $\varepsilon_c=0.02, \varepsilon_s=0.01$ in Table 3, and the updated results are as follows. **Nevertheless, it results in the same conclusion of analysis in the manuscript, and we can confirm that $\varepsilon_c=\varepsilon_s=0.01$ yields the best result.**
>
> | $\varepsilon_c$ / $\varepsilon_s$ (WR / Occurance)| 0.005 | 0.01 | 0.02 |
> |-|:-:|:-:|:-:|
> | 0.005 | 76.4 / 0.072 | 76.7 / 0.073 | 76.4 / 0.074 |
> | 0.01 | 78.4 / 0.244 | **79.2** / 0.245 | 77.4 / 0.240 |
> | 0.02 | 74.9 /0.337 | 74.2 / 0.353 | 74.6 / 0.337 |
> _____
> [1] Scaling Laws for Neural Language Models (Kaplan et al., 2020)
> [2] Iterative Reasoning Preference Optimization (Pang et al., 2024)
> [3] SimPO: Simple Preference Optimization with a Reference-Free Reward (Meng et al., 2024)

---

> ### Author Response · Authors · 2025-08-07
> **Official Comment by Authors**
>
> Dear Reviewer gyoi,
>
> This is a gentle reminder that the original Reviewer-Author Discussion period has been extended to August 8 (AoE), allowing for further discussion. We have tried our best to address the five key points of discussion you provided in our rebuttal: **1. Conditions for Reliable Approximation [W1], 2. Analysis of Additional Computation Cost [W2, Q3], 3. Performance Differences of
> $\varepsilon$-DPO across Base Models [W3, Q2], 4. Alternative for the Updating Rule of $\beta$
>  [W4], 5. Clarification of the Notation and Experiments [Q1, Q4]**. If you still have any further questions, we are eager to address them and would appreciate your efforts during the Reviewer-Author Discussion period.

---

### Official Review · Reviewer_xKby · 2025-07-05

**Clarity:** 4
**Significance:** 2
**Originality:** 2
**Rating:** 4
**Confidence:** 3

**Summary:**

As an RLHF algorithm, DPO has its limitations, in that it uses a constant KL penalty coefficient β, which may lead to suboptimal performance.

To address this, ε-DPO proposes an instance-level adaptive KL penalty control mechanism. It follows the direction of β that brings better separability of winner samples and loser samples ($z_\theta\left(x, y^w, y^l\right):=\log \frac{\pi_\theta\left(y^w \mid x\right)}{\pi_\theta\left(y^l \mid x\right)}$). However, to estimate the separability, one has to be able to estimate the solution of DPO $\theta(\beta)$ for each $\beta$, which is not possible.

To address this, ε-DPO utilizes the estimation from Liu et al., to model the policy obtained by $\theta(\beta/\gamma)$ by an interpolation of  the policy obtained by $\theta(beta)$ and the reference policy. Therefore, ε-DPO updates $\beta$ at each iteration.

Experiments on benchmarks such as AlpacaEval 2, Arena-Hard, and MT-Bench demonstrate that ε-DPO outperforms DPO and several other direct alignment methods in LC win rate.

**Questions:**

1. How many epochs do you train, for each baselines, and your algorithms? Do you use the same epoch number?

2. How do you set the initial $\beta$ value, do you have to tune $\beta$ and $\epsilon$ at the same time?

**Ethical Concerns:**

["NO or VERY MINOR ethics concerns only"]

**Final Justification:**

My main concern focuses on sensitivity, computational cost, robustness of perturbed policies, and lack of experimental details. The authors have provided detailed explanation and statistics regarding each of my concern, and therefore I decided to keep my score voting for acceptance.

**Limitations:**

The authors discussed the limitations in the last paragraph.

**Quality:**

3

**Strengths And Weaknesses:**

Strength:
1. The paper is clearly written.

2. The paper utilizes an effective way of estimating the expected change in separability under a small change in KL penalty coefficient β.

3. The paper clearly discussed its relationship with prior related work $\beta$-DPO and TR-DPO, with performance comparison.

4. The performance on  Llama-3-Instruct (8B) Arena-Hard is good, showing high potential.

Weakness:

1. The performance in Table 1 is not pronounced. I see that the performance based on Mistral has lower WR on AlpacaEval, ArenaHard, and the performance based on Llama3-Instrcut has a lower MT-Bench score.

2. While I understand that the performance may not outperform on every benchmark and every dataset, I suspect that this might be due to the sensitivity of $\epsilon$, when changing datasets.

3. The algorithm requires additional computation overhead, and the training time is not compared.

4. Intuitively, the result of one step of optimizing $\theta$ might be far from the estimation in this paper, leading to an inaccurate estimation of the direction, the accuracy of this estimation, and the accuracy of the direction of $\beta$ has not been discussed.

5. The performance comparison with $\beta$-DPO and TR-DPO on base model Mistral has not been shown.

---

> ### Author Rebuttal · Authors · 2025-07-31
>
> **1. Consideration about Sensitivity of $\varepsilon$ to Performance [W1, W2]**
>
> > **[W1]** The performance in Table 1 is not pronounced. I see that the performance based on Mistral has lower WR on AlpacaEval, ArenaHard, and the performance based on Llama3-Instrcut has a lower MT-Bench score.
>
> > **[W2]** While I understand that the performance may not outperform on every benchmark and every dataset, I suspect that this might be due to the sensitivity of $\epsilon$, when changing datasets.
>
> We sincerely appreciate your careful observations of the experimental results. We also agree that fluctuation of performance across benchmarks can be affected by the choice of base model, dataset, and hyperparameter search range. **However, we want to emphasize that $\varepsilon$-DPO still significantly improves overall performance, consistent performance improvements were observed in additional experiments on another model, and the sensitivity of $\varepsilon$ is unlikely to be a major factor of performance difference, given that the best model was obtained when $\varepsilon=0.01$ regardless of other conditions.** Due to the limited computation resources, we fix the best method-specific hyperparameters of DPO, SimPO, and $\varepsilon$-DPO obtained in the `Llama-Instruct` setting and trained `Qwen2.5-7B-Instruct` as the base model with a hyperparameter search range of learning rate in [3e-7, 5e-7, 7e-7]. Here, to approximate on-policy learning [1] as `Mistral-Instruct` and `Llama-Instruct`, we constructed preference annotations by `PairRM` as an evaluator from responses sampled from the base model. **The results below show that while SimPO fails under the best method-specific hyperparameters identified in `Llama-Instruct`, $\varepsilon$-DPO achieves concrete performance improvement at the same $\varepsilon$ choice as `Llama-Instruct`.** Therefore, we speculate that $\varepsilon$-DPO's performance on a specific benchmark is likely due to the sensitivity of the base model, rather than sensitivity to the choice of $\varepsilon$.
>
> | Method |AlpacaEval 2 (LC / WR) | Arena-Hard (WR) | MT-Bench (Score) |
> |-|:-:|:-:|:-:|
> | SFT | 27.8 / 27.9 | 51.8 | 8.6 |
> | DPO | 41.6 / **46.3** | 66.8 | 8.9 |
> | SimPO | 32.4 / 46.0 | 60.2 | 8.8 |
> | $\varepsilon$-DPO| **42.5** / 46.1 | **67.5** | **9.1** |
>
> **2. Analysis of Additional Computation Cost [W3]**
>
> > **[W3]** The algorithm requires additional computation overhead, and the training time is not compared.
>
> We sincerely appreciate your comment about the requirement of comparison of the computational cost between $\varepsilon$-DPO and DPO. We agree that clarifying the computational increment can give a more concrete understanding of the strength of $\varepsilon$-DPO. Formally, the estimated forward and backward passes cost per token in FLOPs follow $C_f \approx 2N$ and $C_b \approx 2C_f$ for a given model parameter size $N$, excluding the embedding layer [2]. In the case of DPO, since forward and backward passes for the policy model and forward pass for the reference model occur, the FLOPs per token can be approximated as $8N$. When we approximate the policy model under perturbation of $\beta$, $(2v + v + 5v)$ FLOPs are added per token for a given vocabulary size $v$, which corresponds to two scalar-vector multiplications, vector addition, and log-softmax operation, respectively. **This implies that the relative ratio of additional computation cost in FLOPs per token compared to the computation cost of DPO can be roughly approximated as $\frac{2v}{N}$. Therefore, because $v<<N$ in general, the additional computation cost required by $\varepsilon$-DPO is negligible.** Specifically, `Mistral-7B-Instruct-v0.2` has $v = 32,000$ and `Meta-Llama-3-8B-Instruct` has $v = 128,256$, so the additional computation cost can be estimated to be approximately 0.001% and 0.003%, respectively. To verify whether $\varepsilon$-DPO follows such a small computational cost, we compared the training wall-time increment $\Delta t$ that increased in $\varepsilon$-DPO compared to DPO under `Mistral-Instruct` and `Llama-Instruct` settings. Although there is a difference in the scale compared to the estimate of computational cost, we can verify that $\varepsilon$-DPO still requires a negligible wall-time increment.
>
> | Wall-time Increment | Mistral-Instruct | Llama-Instruct |
> |-|:-:|:-:|
> | Step (Avg) | 0.00082s | 0.00064s |
> | Epoch | 0.38076s | 0.30016s |
> | $\frac{\Delta t}{t} (\\%) $ | 0.0064% | 0.0045% |
>
>
> **3. Understanding the Robustness of Estimated $\beta$-Pertured Policies [W4]**
> > **[W4]** Intuitively, the result of one step of optimizing $\theta$
>  might be far from the estimation in this paper, leading to an inaccurate estimation of the direction, the accuracy of this estimation, and the accuracy of the direction of $\beta$
>  has not been discussed.
>
> We sincerely appreciate your concerns about the robustness of the adaptive control criterion during the training process. We acknowledge that a stable approximation to the optimal policy, which changes as model updates are performed, can significantly influence the quality of controlling $\beta$. The adaptive criterion of $\varepsilon$-DPO assumes following conditions: (1) The current policy can perform sufficient approximation to the optimal policy for a given $\beta$; and (2) When observing logit monotonicity, the logit function $z_{\theta(\beta)}(x, y^w, y^l)$ maintains monotonic order according to $\theta(\beta)$ in the entire neighborhood $(\beta_{\varepsilon}^{-}, \beta_{\varepsilon}^{+})$ for a given $\varepsilon$. To verify how much this condition can be satisfied during the training process, we additionally checked the average upper bound of $\varepsilon$ that define the neighborhood $(-\varepsilon, +\varepsilon)$ that the all value in the neighborhood consistently satisfies the logit monotonic criterion for triplets $(x, y^w, y^l)$ of test splits in Antropich-HH, through checkpoints obtained obtained in 0.1 epoch intervals on DPO with $\beta = 0.05$ and `gemma-2-2b`. **That is, assuming that the approximation of optimal policy for the current $\beta$ strengthens according to training steps, we verify the smoothness of logit monotonicity with respect to $\varepsilon$ by observing the upper bound of $\varepsilon$ that yields consistent adaptive decisions compared to smaller $\varepsilon$.** We tested 100 uniform sample points of $\varepsilon$ within the range (0.005, 0.02). **We observed that the expected upper bound of $\varepsilon$, $\varepsilon_{\downarrow}$ and $\varepsilon_{\uparrow}$, which correspond to the monotonic decreasing and increasing logits, respectively, converged almost at 0.008 after 0.2 epochs, similar to the best result of $\varepsilon=0.01$ of previous experiments, as below.** Furthermore, the lower value of $\varepsilon_{\downarrow}$ and $\varepsilon_{\uparrow}$ at 0.1 epoch is also consistent with the degeneration phenomenon of $\varepsilon$-DPO in the early stages of training discussed in Section 4.3. **Therefore, we can observe that, except for the early stage of training, relatively stable estimations of policy under perturbation of $\beta$ across training steps for adequate selection of $\varepsilon$.**
>
> | Upper Bound of $\varepsilon$ | 0.1 | 0.2 | 0.3 | 0.4 | 0.5 | 0.6 | 0.7 | 0.8 | 0.9 | 1.0|
> |-|-|-|-|-|-|-|-|-|-|-|
> | $\varepsilon_{\downarrow}$ (Avg / Std) | 0.0068 / 0.0011 | 0.0080 / 0.0038 | 0.0081 / 0.0040 | 0.0080 / 0.0040 | 0.0080 / 0.0040 | 0.0084 / 0.0044 | 0.0087 / 0.0047 | 0.0083 / 0.0041 | 0.0083 / 0.0041 | 0.0081 / 0.0038 |
> | $\varepsilon_{\uparrow}$ (Avg / Std) | 0.0067 / 0.0006 | 0.0076 / 0.0032 | 0.0079 / 0.0037 | 0.0079 / 0.0037 | 0.0079 / 0.0038 | 0.0079 / 0.0038 | 0.0084 / 0.0043 | 0.0079 / 0.0043 | 0.0082 / 0.0037 | 0.0083 / 0.0040 |
>
> **4. Clarification of Experimental Details [W5, Q1, Q2]**
> > **[W5]** The performance comparison with
> $\beta$-DPO and TR-DPO on base model Mistral has not been shown.
>
> > **[Q1]** How many epochs do you train, for each baselines, and your algorithms? Do you use the same epoch number?
>
> > **[Q2]** How do you set the initial $\beta$ value, do you have to tune and $\varepsilon$ at the same time?
>
> We sincerely appreciate your careful observation of our experimental settings. **First, we would like to clarify that the comparison of $\beta$-DPO and TR-DPO includes only the `Llama-Instruct` setting because both prior works commonly report in their official papers.** Unfortunately, due to our limited resources, experiments of TR-DPO in the `Mistral-Instruct` setting were unavailable. We speculate that it is due to the memory peak during the reference model parameter update because of gathering parameters to the main process, and also implies the limitation on additional computational overhead and flexibility of TR-DPO that can arise without careful implementation. **As we described in Appendix B, all experimental run in this work is fixed to one epoch. This is to mitigate the over-optimization that commonly occurs in direct alignment algorithms [3], and also SimPO reports that one epoch training generally yields the best performance [1].** Furthermore, due to our limited resources, this work primarily performed a hyperparameter search for $\varepsilon$ while fixing $\beta=0.01$, which is the best parameter commonly found by SimPO in hyperparameter searches for DPO. However, experimental results often show the best performance at $\varepsilon=0.01$. **Therefore, we conjecture that finding an appropriate initial $\beta$ while fixing $\varepsilon=0.01$ would be an efficient search strategy as a general guideline**.
> _____
> [1] SimPO: Simple Preference Optimization with a Reference-Free Reward (Meng et al., 2024)
> [2] Scaling Laws for Neural Language Models (Kaplan et al, 2020)
> [3] Scaling Laws for Reward Model Overoptimization in Direct Alignment Algorithms (Rafailov et al., 2024)

---

> > ### Comment · Reviewer_xKby · 2025-08-09
> >
> > Thank you for the clarifications and additional analyses in your response. I have reviewed your rebuttal and found that it addresses my main concerns regarding sensitivity analysis, computational cost, robustness of perturbed policies, and experimental details. I do not have further questions at this stage and will maintain my current evaluation of acceptance.

---

> ### Author Response · Authors · 2025-08-07
> **Official Comment by Authors**
>
> Dear Reviewer xKby,
>
> This is a gentle reminder that the original Reviewer-Author Discussion period has been extended to August 8 (AoE), allowing for further discussion. We have tried our best to address the four key points of discussion you provided in our rebuttal: **1. Consideration about Sensitivity of $\varepsilon$ to Performance [W1, W2], 2. Analysis of Additional Computation Cost [W3], 3. Understanding the Robustness of Estimated $\beta$-Pertured Policies [W4], 4. Clarification of Experimental Details [W5, Q1, Q2]**. If you still have any further questions, we are eager to address them and would appreciate your efforts during the Reviewer-Author Discussion period.

---

### Official Review · Reviewer_c65B · 2025-07-06

**Clarity:** 3
**Significance:** 4
**Originality:** 2
**Rating:** 5
**Confidence:** 4

**Summary:**

This paper introduces ε-DPO, an instance-level adaptive extension to Direct Preference Optimization (DPO) for aligning large language models with human preferences in offline setting. Instead of static KL penalty in DPO, this paper proposes a novel mechanism that is adaptive to each preference pair. This gives more informative KL divergence info, because of the monotonicity of the log-likelihood ratio between preferred and rejected responses.

**Questions:**

Q1. The monotonicity of the log-likelihood ratio holds assuming scalar evaluation. Does the intransitivity observed in real datasets at the instance and annotator levels have any potential impact or connection to the proposed method? How does this relate to the discussed 'confusion on preference pairs'?

Q2. How to elaborate the additional computational cost in the proposed adaptive instance-level operations, when compared with batch operations?

**Ethical Concerns:**

["NO or VERY MINOR ethics concerns only"]

**Final Justification:**

I will recommend as 'Accept'.

The authors were able to provide their dedicated response with reminiscence. It is essential to highlight the monotonicity property, that the logit monotonicity criterion of $\varepsilon$-DPO is derived from the properties of DPO, whose assumption of the preference model is the Bradley-Terry model.

Also, it is now crystal clear regarding the computational considerations.

**Limitations:**

yes

**Paper Formatting Concerns:**

No issue is identified regarding the submitted paper content, paper references, the NeurIPS paper checklist.

**Quality:**

4

**Strengths And Weaknesses:**

Strength:
1. Instance-level Adaptive KL Control:
Proposes ε-DPO, which replaces DPO’s static KL penalty with a simple yet effective adaptive mechanism at the instance level, achieving superior performance on general chatbot benchmarks.

2. Preference Confidence–Driven Adjustment:
Reinterprets DPO’s preference modeling framework as a basis for dynamic KL control, by estimating how preference confidence changes under small perturbations of the KL coefficient β.

3. Experimental results are comprehensive and promising:
Experiments on UltraFeedback and Anthropic-HH datasets demonstrated the good performance of the proposed method in the majority of the trials.

Weakness:
1. Although the proposal is theoretically justified from the monotonicity of the log-likelihood ratio, the assumption is dependent on the transitivity of the reward function, which might not hold in a real dataset given human annotations.
- The literature below studied representative preference datasets in the real world, where the 'transitive' relationship between preference annotations may not always hold.
- https://arxiv.org/abs/2409.19325 (Duan et al, 2017)

2. Previous works have inefficiencies in computational cost, as described in Lines 114-116; however, this paper does not discuss this aspect with evidence in its proposed algorithms and experiments.

---

> ### Author Rebuttal · Authors · 2025-07-31
>
> **1. Effect of Intransitivity to Logit Monotonicity [W1, Q1]**
>
> > **[W1]** Although the proposal is theoretically justified from the monotonicity of the log-likelihood ratio, the assumption is dependent on the transitivity of the reward function, which might not hold in a real dataset given human annotations. The literature below [1] studied representative preference datasets in the real world, where the 'transitive' relationship between preference annotations may not always hold.
>
> > **[Q1]** The monotonicity of the log-likelihood ratio holds assuming scalar evaluation. Does the intransitivity observed in real datasets at the instance and annotator levels have any potential impact or connection to the proposed method? How does this relate to the discussed 'confusion on preference pairs'?
>
> We sincerely appreciate your thoughtful consideration of the theoretical justification of $\varepsilon$-DPO. We also agree that the adaptive control criterion of $\varepsilon$-DPO, logit monotonicity, relies on the assumption of scalar-valued and transitivity of the ground-truth reward function. **However, we want to clarify that this assumption starts from the Bradley-Terry model of DPO rather than our new assumption as an extension of DPO**. Specifically, $\varepsilon$-DPO, like DPO, assumes the Bradley-Terry model, which implies the ground-truth reward function $r^*(x, y)$ can impose a total ordering of response candidates $y$ for prompt $x$ through reward value difference by the reward value difference of two candidates. Also, if we treat the observation triplet $(x, y^w, y^l)$ represents the hard label in ground-truth preference model (i.e., $\mathbb{P}(y^w \succ p^l|x) = 1$), we can assume that an estimated preference model that assigns higher confidence to the preference pair serve as more accurate estimation, which is the way we control the $\beta$, corresponds to the training-time temperature of the estimated preference model. **In other words, the logit monotonicity criterion of $\varepsilon$-DPO is derived from the properties of DPO, whose assumption of the preference model is the Bradley-Terry model.**
>
> Therefore, intransitivity, which is often observed in real-world human preference annotations, violates not only the $\varepsilon$-DPO but also the DPO's assumption that scalar-value score differences can impose total ordering. **This implies that the possibility of intransitivity in preference annotation implies the limitation not only in the robustness of $\varepsilon$-DPO's adaptive criterion but also in the DPO loss objective itself.** Suppose we consider the 'confusion on preference pairs' problem from the perspective of this violation of transitivity. If the training dataset contains duplicated responses y as a training instance of a preference triplet $(x, y^w, y^l)$ for a given $x$, intransitivity might exist between them. **In this specific case, perfect classification via the scalar-valued reward difference is impossible. However, unlike DPO, $\varepsilon$-DPO can indirectly adjust the training-time inverse temperature $\beta$ for these samples, which might mitigate this inseparability problem caused by the Bradley-Terry model by adjusting the preference confidence.**
>
> Still, $\varepsilon$-DPO relies on the Bradley-Terry model assumption as DPO and cannot completely mitigate intransitivity. Therefore, training on a dataset with a strong suspicion of intransitivity could significantly impact the final model performance. To address this issue, adopting the perspective of KTO [2], which incorporates the Kahneman-Tversky model that can accommodate the intransitivity, could be an effective alternative. **However, unlike DPO, which can easily estimate the logit monotonicity based on perturbations to $\beta$, it is challenging to express the optimal policy under the KTO loss objective in closed form.** Also, the fact that KTO did not show a significant performance improvement compared to DPO in Table 1 suggests that the negative impact of intransitivity on the final model performance in a well-constructed dataset may not be significant.
>
> **2. Analysis of Additional Computation Cost [W2, Q2]**
>
> > **[W2]** Previous works have inefficiencies in computational cost, as described in Lines 114-116; however, this paper does not discuss this aspect with evidence in its proposed algorithms and experiments.
>
> > **[Q2]** How to elaborate the additional computational cost in the proposed adaptive instance-level operations, when compared with batch operations?
>
> We sincerely appreciate your comment about the missing discussion of the additional computational cost induced by $\varepsilon$-DPO. We agree that a specific discussion about the additional computational cost of $\varepsilon$-DPO is meaningful for the claim of this work. Formally, the estimated forward and backward passes cost per token in FLOPs follow $C_f \approx 2N$ and $C_b \approx 2C_f$ for a given model parameter size $N$, excluding the embedding layer [3]. In the case of DPO, since forward and backward passes for the policy model and forward pass for the reference model occur, the FLOPs per token can be approximated as $8N$. When we approximate the policy model under perturbation of $\beta$, $(2v + v + 5v)$ FLOPs are added per token for a given vocabulary size $v$, which corresponds to two scalar-vector multiplications, vector addition, and log-softmax operation, respectively. **This implies that the relative ratio of additional computation cost in FLOPs per token compared to the computation cost of DPO can be roughly approximated as $\frac{2v}{N}$. Therefore, because $v<<N$ in general, the additional computation cost required by $\varepsilon$-DPO is negligible.** Specifically, `Mistral-7B-Instruct-v0.2` has $v = 32,000$ and `Meta-Llama-3-8B-Instruct` has $v = 128,256$, so the additional computation cost can be estimated to be approximately 0.001% and 0.003%, respectively. To verify whether $\varepsilon$-DPO follows such a small computational cost, we compared the training wall-time increment $\Delta t$ that increased in $\varepsilon$-DPO compared to DPO under `Mistral-Instruct` and `Llama-Instruct` settings as follows. Although there is a difference in the scale compared to the estimate of computational cost, we can verify that $\varepsilon$-DPO still requires a negligible wall-time increment.
>
> | Wall-time Increment | Mistral-Instruct | Llama-Instruct |
> |-|:-:|:-:|
> | Step (Avg) | 0.00082s | 0.00064s |
> | Epoch | 0.38076s | 0.30016s |
> | $\frac{\Delta t}{t} (\\%) $ | 0.0064% | 0.0045% |
>
> Although the additional computation cost incurred by $\varepsilon$-DPO is negligible, it is proportional to the vocabulary size $v$ compared to the batch-level operation of $\beta$-DPO that performs an average operation of implicit rewards. **However, $\beta$-DPO inevitably is affected by the size of the mini-batch in terms of the accuracy of estimating the statistics of the implicit reward, which causes difficulties in utilizing techniques such as parallelism and gradient accumulation.** Under the data parallelism, $\beta$-DPO must perform all-gather and all-scatter operations of the implicit reward obtained for each process to fully obtain the statistics of the implicit reward for the entire mini-batch. Moreover, because it is necessary to obtain the $\beta$ to be used for each micro-batch when performing gradient accumulation, we can only use the statistics from the micro-batch size before the actual optimizer update step is performed. **On the other hand, it is noteworthy that $\varepsilon$-DPO is not only free from the constraint of mini-batch size, but also requires negligible additional computation cost for instance-level adaptive control of $\beta$, which $\beta$-DPO cannot achieve.**
> _____
> [1] A Generalized Model for Multidimensional Intransitivity (Duan et al., 2017)
> [2] KTO: Model Alignment as Prospect Theoretic Optimization (Ethayarajh et al., 2024)
> [3] Scaling Laws for Neural Language Models (Kaplan et al., 2020)

---

> > ### Comment · Reviewer_c65B · 2025-08-03
> >
> > I thank the authors for providing their dedicated response with reminiscence. Monotonicity is at the heart of reward modeling; it is indeed essential to highlight that the logit monotonicity criterion of $\varepsilon$-DPO is derived from the properties of DPO, whose assumption of the preference model is the Bradley-Terry model. Also, it is now crystal clear regarding the computational considerations. I will maintain my score as 'Accept'.

---

### Comment · Area_Chair_sWp9 · 2025-08-05

Dear Reviewers,

This is a gentle reminder that the authors have responded to your question. Could you please provide your feedback on their response?

Kindly note that the deadline of August 6 is approaching, and your timely feedback would be greatly appreciated to facilitate any further discussions if needed.

---

### Note · Authors · 2025-08-13

We sincerely appreciate thoughtful comments and valuable feedback from all reviewers. We want to highlight that all reviewers **[c65B, xKby, gyoi, 4tbE]** remarked that our approach, $\varepsilon$-DPO, effectively addresses the static KL penalty, a key bottleneck of DPO, while achieving instance-level adaptive control, which existing approaches fail to achieve. Furthermore, the reviewers find that this simple approach of estimating the advantage of perturbation on $\beta$ by checking preference confidence variation based on the DPO perspective **[c65B, xKby, 4tbE]** demonstrated empirically promising performance **[c65B, xKby, gyoi]**, supporting the efficacy of $\varepsilon$-DPO. We find that some commonalities of concern exist across reviewers: 1. Additional Computational Cost of $\varepsilon$-DPO **[c65B, xKby, gyoi]**, 2. Sensitivity of $\varepsilon$-DPO across Experimental Settings **[xKby, gyoi]**, 3. Robustness on Estimating $\beta$-perturbed policy **[xKby, gyoi]**, and we made efforts to address each reviewer's comments during the rebuttal period. We are pleased that all reviewers **[c65B, xKby, gyoi, 4tbE]** responded that our rebuttal addressed their major concerns during the Reviewer-Author Discussion period. We are eager to incorporate these discussions and suggestions from all reviewers in the final version for further improvement.

---

### Decision · Program_Chairs · 2025-09-17

**Decision:**

Accept (poster)

**Comment:**

This paper proposes ϵ-DPO, an instance-level adaptive extension of Direct Preference Optimization (DPO) for aligning large language models with human preferences. Instead of a fixed KL penalty, ϵ-DPO dynamically adjusts the KL term per preference pair by checking the monotonicity of log-likelihood ratios under small β perturbations. This approach balances adherence to the reference policy with alignment to preference data.

Strengths:
- Principled instance-level KL adaptation addressing a key limitation of DPO.
- Strong empirical performance on diverse conversational benchmarks.
- Theoretical justification based on logit monotonicity and Bradley-Terry preference modeling.
- Clear writing and well-situated in prior literature.

After rebuttal, reviewers generally agreed that concerns were addressed: computational feasibility is clarified, approximation errors are mitigated, and the practical benefits on conversational benchmarks are evident. As such, the paper merits acceptance.